# Environmental and climate variability drive population size of annual penaeid shrimp in a large lagoonal estuary

Lela S. Schlenker[1]*, Chris Stewart[2], Jason Rock[2], Nadine Heck[3], James W. Morley[1]

**1** Department of Biology, Coastal Studies Institute, East Carolina University, Wanchese, North Carolina, United States of America, **2** Department of Environmental Quality, North Carolina Division of Marine Fisheries, Morehead City, North Carolina, United States of America, **3** Department of Coastal Studies, Coastal Studies Institute, East Carolina University, Wanchese, North Carolina, United States of America

\* schlenkerl20@ecu.edu

**Data Availability Statement:** Raw data are publicly available from http://www.seamap.org/. Processed data used in models are available at https://doi.org/10.5061/dryad.m37pvmd76.

## Abstract

Species with short life spans frequently show a close relationship between population abundance and environmental variation making these organisms potential indicator species of climatic variability. White (*Penaeus setiferus*), brown (*P. aztecus*), and pink (*P. duorarum*) penaeid shrimp typically have an annual life history and are of enormous ecological, cultural, and economic value to the southeastern United States and Gulf of Mexico. Within North Carolina, all three species rely on the Pamlico Sound, a large estuarine system that straddles Cape Hatteras, one of the most significant climate and biogeographic breaks in the world, as a nursery area. These characteristics make penaeid species within the Pamlico Sound a critical species-habitat complex for assessing climate impacts on fisheries. However, a comprehensive analysis of the influence of the environmental conditions that influence penaeid shrimp populations has been lacking in North Carolina. In this study, we used more than 30 years of data from two fishery-independent trawl surveys in the Pamlico Sound to examine the spatial distribution and abundance of adult brown, white, and pink shrimp and the environmental drivers associated with adult shrimp abundance and juvenile brown shrimp recruitment using numerical models. Brown shrimp recruitment models demonstrate that years with higher temperature, salinity, offshore windstress, and North Atlantic Oscillation phase predict increased abundance of juveniles. Additionally, models predicting adult brown, white, and pink shrimp abundance illustrate the importance of winter temperatures, windstress, salinity, the North Atlantic Oscillation index, and the abundance of spawning adult populations from the previous year on shrimp abundance. Our findings show a high degree of variability in shrimp abundance is explained by climate and environmental variation and indicate the importance of understanding these relationships in order to predict the impact of climate variability within ecosystems and develop climate-based adaptive management strategies for marine populations.

**Funding:** This research was supported by a grant from North Carolina Sea Grant (NCSG-RM-02) to JWM and NH. The funders had no role in study design, data collection and analysis, decision to publish, or preparation of the manuscript.

**Competing interests:** The authors have declared that no competing interests exist.

## Introduction

Human driven climate change is causing the redistribution of species around the globe [1]. In marine systems, over 80% of all observations for changes in the distribution, phenology, ecosystem structure, and abundance of marine organisms are already consistent with predictions for ocean warming and show geographic shifts poleward and to deeper waters [2–5]. Further, impacts to marine ecosystems are expected to intensify regardless of carbon mitigation strategies [6] making it essential to understand the relationships between climate variables and regional population productivity. This is because changes in regional productivity are considered a major driver of distribution shifts [7].

Less is known about how climate change will impact estuarine systems as compared to the coastal ocean, but it is anticipated that complex shifts in estuarine conditions will occur and alterations in habitat use by estuarine species have already been observed [8,9]. The Pamlico Sound in North Carolina is the second largest estuary in the United States and straddles Cape Hatteras, one of the most significant climate and biogeographic breaks in the world [10,11]. This unique shallow ecosystem is further defined by its primarily wind driven circulation and limited ocean connectivity [12,13]. Like many coastal regions, both the climatology and the biological community of the Pamlico Sound are changing. Among these changes are increased salinity caused by rising sea levels [14], elevated temperatures [15], shifts in species composition [15], eutrophication [16], and increased storm frequency [17]. Additionally, commercial shrimp trawling continues to heavily impact the Pamlico Sound and coastal shelf food webs [18,19].

The Pamlico Sound is also the most productive penaeid shrimp nursery in the southeast U. S. [20,21], making it a critical region to examine the impacts of climate on fisheries. The commercial shrimp fishery in this region is supported by three penaeid shrimp species: brown (*Penaeus aztecus*; Ives, 1891), white (*P. setiferus*; Linnaeus, 1767), and pink (*P. duorarum*; Burkenroad, 1939). Shrimp are routinely the most valuable fishery in North Carolina, and in 2020, the most recent year for which data are available, the landings value of the shrimp harvest was $22.3 million [22]. Shrimp are also ecologically important as a source of prey for invertebrate and finfish predators, in turn supporting commercially and recreationally important fisheries as they migrate annually between the Pamlico Sound and the continental shelf [23–25].

Due in part to the complex annual life cycle of penaeid shrimp species, in which adults reproduce in ocean waters and postlarval shrimp recruit to estuarine waters, variation in both localized and broad scale environmental and climate variables are expected to have an impact on recruitment success, abundance, and distribution of penaeid shrimp populations. For many marine species the relationship between the number of spawning adults and offspring is weak, and recruitment is thought to be strongly influenced by environmental conditions [26–28]. However, as a result of their annual life cycle, penaeid species have the potential to be a model organism to examine the relationship between adult spawners and successful recruits. Understanding the relationship between the distribution of spawning adults and subsequent juvenile recruitment has become critically important for species with complex ontogenetic shifts in habitat requirements. For example, climate change may shift movement patterns of spawning adults, which may impact spatial recruitment patterns of estuarine-dependent juveniles [7,8]. For penaeid species, the spawner-recruit relationship together with environmental and climate variables, may be a useful metric for predicting population abundance and bioclimatic variability [29] and contribute to informed management decisions [30].

Despite the the economic and ecological importance of penaeid shrimp, and the changing environment of the Pamlico Sound, a comprehensive analysis of the impact of climate and environmental variability on penaeid species has been lacking. To address that gap, the present study used more than 30 years of fishery-independent data of juvenile brown shrimp, and sub-

adult and adult brown, white, and pink shrimp in the Pamlico Sound with environmental data to examine: 1) the spatial distribution and abundance of sub-adult and adult penaeid shrimp and 2) the environmental variables that influence the recruitment success of juvenile brown shrimp and the abundance of sub-adult and adult brown, white, and pink shrimp. Our results demonstrate that annual variability in penaeid shrimp populations within the Pamlico Sound are highly influenced by environmental conditions and regional spawning stock biomass. Therefore, these relationships may provide a useful framework for understanding how future climate shifts may impact penaeid shrimp species in North Carolina and the U.S. east coast as well as providing insight into the rate of bioclimatic variability.

## Methods

### Biological survey data

Catch data for juvenile brown shrimp was collected by the North Carolina Division of Marine Fisheries (NCDMF) for their Estuarine Trawl Survey (Program 120, hereafter referred to as "P120" or the "spring" survey) and therefore did not require sampling permits [31]. The P120 survey is a standardized trawl survey (one-minute tows) that occurs in the tidal creeks of the Pamlico Sound, including the lower reaches of all major river systems emptying into the Pamlico Sound, and also samples regions in central and southern North Carolina (Fig 1). The P120 survey started in 1971 with the goal of producing recruitment indices within juvenile nursery habitat for several species including brown shrimp. Sampling for the survey occurs during the middle two weeks of May and June (Table 1) and the timing is structured to capture peak abundance of juvenile brown shrimp within the tidal creeks. While the sampling does capture some pink and white shrimp, which spawn later in the year than brown shrimp, the timing of the survey is not appropriate for assessing juvenile abundance of those species [32,33]. Therefore, our analyses of the P120 survey data used only juvenile brown shrimp data. Effort in the survey has varied over time so we opted to use data only from 102 fixed core stations that are sampled in May and again in June within the Pamlico Sound system. Additionally, in order to match available climate data, only data from 1986 to 2019 were used.

Catch data for adult brown, white, and pink shrimp was collected by NCDMF for their Pamlico Sound Trawl Survey (Program 195, hereafter referred to as "P195" or the "summer and fall" survey) and did not require sampling permits [31]. The P195 trawl survey began in 1987 and occurs each June (summer) and September (fall; mean = 52.5 hauls per season; Table 1). The survey employs 20-minute tows in a stratified random design based on depth each season and year and occurs in the main body ($> 2$ m) of the Pamlico Sound (Fig 1). The design of the P195 survey is intended to intercept shrimp as they emigrate from shallow nursery habitats and pass through the sound (mean depth ~4.8 m), on their way to coastal inlets. Our analyses of P195 data from 1987 to 2019 used the abundance of adult brown and pink shrimp data from both seasons (June and September) in season-specific models, and based on extremely low catch rates of white shrimp in the summer, the abundance of white shrimp caught only in September. Although both the second sampling of P120 sites and the summer P195 survey both occur in June, the relatively long brown shrimp spawning window and the rapid growth of juvenile brown shrimp means that both surveys are well timed to capture the same cohort of brown shrimp. Additionally, due to survey timing, the P195 survey catches both adult and sub-adult shrimp of brown, white, and pink shrimp (S1 Fig), but hereafter, for our purposes, shrimp caught in this survey will be referred to as "adult." For both the P120 and P195 surveys, trawl catch by species was expressed as the number of shrimp per hectare (catch per unit effort; CPUE) and the dataset was expanded to include zeros for any site (i.e., location where trawling was completed) that did not include catch of a particular species.

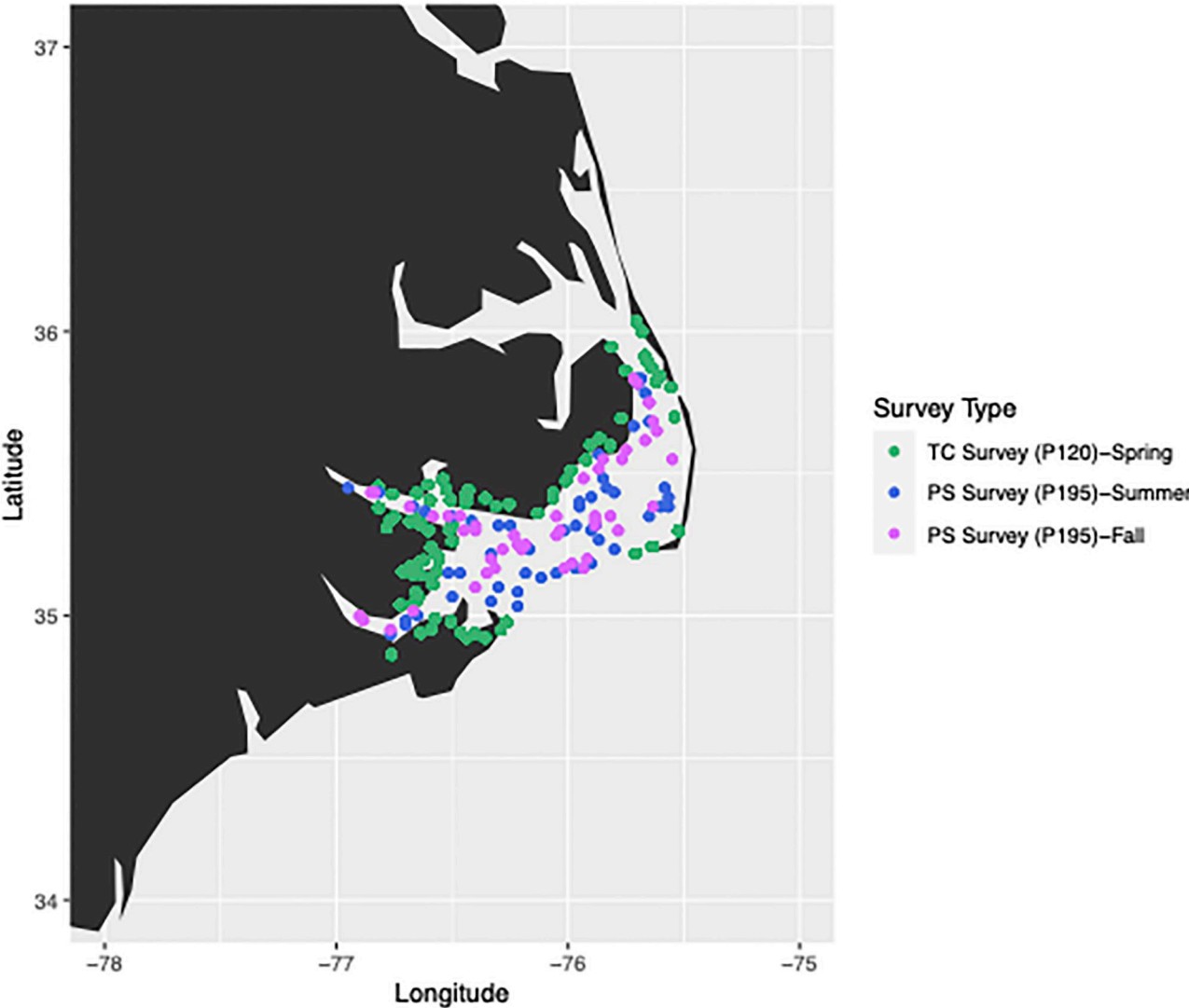

**Fig 1. An example from 2019 of the tidal creek (TC; P120) spring recruitment and Pamlico Sound (PS; P195) adult shrimp survey sites.** The same tidal creek sites are sampled each year while Pamlico Sound adult shrimp sampling sites follow a stratified random design based on depth each season and year.

### Spawning stock indices

Fisheries-independent and -dependent indices of spawning stock abundance were developed for each shrimp species and included as predictors for models estimating the abundance of shrimp in Pamlico Sound. The fishery-independent index used the South Carolina

**Table 1. Species life history and survey data information.** The spawning and recruitment period was determined based on information from Pamlico Sound collected by Williams (1955) [32].

| Species and Life Stage | Survey Data Used | Data Time Period | Habitat Sampled | Survey Timing | Spawning and Recruitment Period |
|---|---|---|---|---|---|
| Juvenile brown shrimp | Program 120 | 1986–2019 | Tidal Creeks | May and June | January-April |
| Adult brown shrimp | Program 195 | 1987–2019 | Pamlico Sound (>2 m) | June and September | January-April |
| Adult white shrimp | Program 195 | 1987–2019 | Pamlico Sound (>2 m) | September | May-July |
| Adult pink shrimp | Program 195 | 1987–2019 | Pamlico Sound (>2 m) | June and September | June-August |

Department of Natural Resources Southeast Area Monitoring and Assessment Program (SEA-MAP), which operates a nearshore trawl survey in spring, summer, and fall and samples between Cape Hatteras, North Carolina and Cape Canaveral, Florida. For each year and season (spring: mid-April to mid-May, summer: mid-July to early August, and fall: early October to mid-November), the mean log CPUE (individuals per hectare) of adult brown, white, or pink shrimp caught on North Carolina's continental shelf was calculated using the SEAMAP trawl survey data [31]. The North Carolina specific index was calculated by averaging the species-specific CPUE for each of the six nearshore SEAMAP strata located between Cape Fear and Cape Hatteras (i.e., the Onslow and Raleigh Bay regions) before determining the mean annual log CPUE across North Carolina strata within representative seasons. The seasonal indices evaluated in models were selected based on the spawning period for each species [32]; the fall survey lagged one year was used for brown shrimp, the spring survey was used for white shrimp, and both the fall survey lagged one year and the spring survey were evaluated for pink shrimp. For our fishery-dependent index of spawning adults, commercial catch data were obtained from the NCDMF and annual species-specific commercial catch indices (standard-ized pounds of shrimp, i.e., a mean of 0 and a standard deviation of 1) from within the Pamlico Sound were calculated and lagged one year.

## Environmental data

Both the P120 and P195 surveys collect *in situ* temperature and salinity data from each sampled site and these environmental variables are represented for the entire timeline of the datasets. Wind speed, wind direction, and water temperature data representing the presumed spawning and pelagic larval duration time period on the continental shelf were gathered from the National Data Buoy Center [34]. Buoys (FPSN7, 41013, DSLN7, 41025, CHLV2, and 44099) were selected based on the following criteria: data covered the study period (1986–2019), location was in close proximity to the Pamlico Sound, and measurements included water temperature, wind speed, and wind direction. Wind speed and wind direction data were collected from all buoys while, as a result of Gulf Stream water frequently influencing temperature readings at several of the buoys, water temperature (winter and spring) was averaged only from CHLV2 and 44099, both of which showed no evidence of Gulf Stream interruption.

For all buoys, daily buoy-specific averages were calculated for the variables of interest before determining daily averages across buoys. From these daily averages, mean water temperature was calculated for winter (December 21-March 20) and for spring until the start of the P120 survey (March 21-May 14). North-south and east-west (oceanographic convention) winds-tresses were calculated using wind speed and direction using the equations from Munch and Conover [35]. Mean daily east-west ($\tau e$) and north-south ($\tau n$) windstress was summed for the species-specific spawning and recruitment periods identified by Williams (Table 1) [32]. Additionally, given the potential for wind to impact the distribution of shrimp at specific stages of their ontogeny, we used the *ClimWin* package, an approach that uses a sliding window to compare the effect of various time periods of an environmental variable on a chosen response variable (in this case, estimated adult shrimp abundance, see below for details), to identify more specific periods of recruitment impact in our windstress data [36]. This approach highlighted north windstress in the month of April and the previous December as potentially important predictor variables for the estimated abundance of summer brown shrimp and fall white shrimp, respectively, and these windstress values from specific time periods were calculated as described for the spawning and recruitment periods and evaluated in model fitting. This approach did not identify a period of windstress importance for pink shrimp outside of the spawning and recruitment window.

River volume data from United States Geological Survey (USGS) stream gauges were used as a proxy for precipitation influence at the watershed scale. Two stream gauges were selected, gauge 02082770, located on Swift Creek in Hilliardston, North Carolina, and gauge 02083000, located on Fishing Creek in Enfield, North Carolina. These two stream gauges met the criteria for data that covered the time period of the study (1986–2019), were located on creeks that fed into the Pamlico Sound, and were not dammed. Monthly stream flow data in cubic feet per second were used to calculate the mean annual flow (up to the date the survey began) and the mean flow during the spawning and recruitment period were calculated and standardized to a mean of 0 and a standard deviation of 1 for use in models. Finally, the mean North Atlantic Oscillation (NAO), the principal component-based sea level pressure anomalies over the Atlantic sector, was calculated both annually and for the species-specific spawning and recruitment time period, and these variables were used as potential predictor variables for abundance [37].

## Spatial distribution modeling of brown, white, and pink shrimp

Generalized additive models (GAMs) were used to evaluate spatiotemporal patterns in annual abundance of adult brown, white, and pink shrimp within the Pamlico Sound. The log CPUE values from the P195 summer and fall survey were used in season specific GAMs with year and haul coordinates (longitude and latitude) as predictor variables. These GAMs were used as an approach to account for the spatial autocorrelation inherent in survey data. It should be noted that we used this approach with the P195 survey, which occurs across a continuous section of the Pamlico Sound, but, due to the nature of the P120 sampling, which occurs in discrete tidal creeks spread throughout the Pamlico Sound, we did not consider spatial GAMs appropriate for the P120 juvenile brown shrimp data. GAMs were constructed using the *mgcv* package [38] in R [39] using the Tweedie error distribution and a gamma penalty of 1.4 [40]. The Tweedie error distribution is particularly useful for zero-inflated data (such as survey data) and allows for the use of a single model to estimate abundance. The gamma penalty of 1.4 forces each model effective degrees of freedom to count as 1.4 degrees of freedom which increases smoothness and reduces overfitting [40,41]. Four model types were evaluated:

1. logCPUE~$te$(longitude, latitude, by = year)

2. logCPUE~$s$(longitude, latitude, $k$ = 50)+ year +$ti$(longitude, latitude, by = year)

3. logCPUE~$s$(longitude, latitude, $k$ = 100)+ year +$ti$(longitude, latitude, by = year)

4. logCPUE~$s$(longitude, latitude, $k$ = 50)+$s$(year numeric, $k$ = 30) +$ti$(longitude, latitude, year numeric, $d$ = c(2,1), $k$ = c(50,30)))

Where *s* represents a smoothing parameter appropriate for variables measured in the same units, *te* and *ti* are smoothing parameters for quantities measured in different units, *k* represents the maximum degrees of freedom allowed for each term, and *d* allows you to specify which parameters receive a particular *k* value [40]. Model 1 allowed spatial abundance to vary across years with optimal model complexity (i.e., *k*) determined during the model fitting. Model types 2–4 included a spatial term *s* (longitude, latitude) independent from temporal variation with two different levels of complexity (k = 50 or 100). The inclusion of a separate spatial term was found to reduce unrealistic CPUE prediction values in certain years, particularly in areas along the edge of the prediction grid. Model types 1–3 considered year as a factor, while model type 4 evaluated year as a numeric variable that modeled abundance trends across years. A model for each species and season was selected using Akaike's Information Criterion (AIC) [42,43]. GAM predictions from the selected models were exponentiated and scaled to a

15 arc second grid in order to project predictions onto a spatial grid of the survey area within the Pamlico Sound using the General Bathymetric Chart of the Oceans (GEBCO) dataset and grid cells <2m in depth were removed to appropriately represent the P195 sampling area [44]. To reduce unrealistic predictions that rarely occurred at the edges of the projection grid (i.e., spatial extrapolation error based on a nearby high observed catch), predictions that exceeded the maximum observed catch for each species, season, and year were replaced by the respective maximum observed catch value for that specific species, season, and year. Predictions were summed across grid cells to calculate a predicted annual number for each shrimp species within the Pamlico Sound survey area. This annual estimate was used as the modeled response variable for evaluating environmental predictors of adult shrimp abundance (see below) and for identifying important periods of windstress outside of the spawning and recruitment window with the *ClimWin* package [36] (see above).

## Abundance modeling of juvenile brown and adult brown, white, and pink shrimp

To examine the relationship between environmental variables and juvenile brown shrimp recruitment, the mean annual log CPUE from the P120 survey was modeled with the suite of predictor variables described above (i.e., environmental and spawning stock indices) using generalized linear models (GLMs). Because the P120 survey is conducted in May and June (Table 1), we evaluated the CPUE data from both months for use in predictive models. The CPUE from June was greater and much more consistent over the timeseries than the CPUE from May. For that reason, we opted to develop these models using only the log June CPUE.

The predicted annual number of adult brown, white, and pink shrimp in the Pamlico Sound generated in the spatial GAMs was used in species and season specific GLMs to examine the relationship between predicted annual numbers of adult shrimp and environmental variables and estimates for spawning stock biomass. Adult pink shrimp during summer represented a unique case, because length distribution data showed that these were larger age-1 individuals that were spawned the previous year (S1 Fig). Pink shrimp juvenile recruitment takes place during summer [32], and subadults overwinter in the estuary [45,46]. Therefore, lagged variables (i.e., from the previous year) were evaluated for predicting summer pink shrimp abundance. For adult brown shrimp during summer and fall, the mean log CPUE of juveniles from May of the same year (i.e., P120 survey) was also included as a potential predictor variable. The inclusion of the May P120 CPUE evaluates whether year-classes of brown shrimp are stable after initial recruitment to nursery habitats. The June catches from the P120 survey were excluded due to the overlap of June P120 and P195 sampling.

All data analysis was completed in R [39]. Data visualization and mapping was completed using the *tidyverse* package [47]. A value of 1 was added to catches and estimated abundances before values were logged to adjust for values of zero. Quadratic terms were evaluated for use in GLMs for variables where data visualization suggested nonlinear relationships. Collinearity of terms was evaluated using the *car* package [48] during model fitting and variables with a variance inflation factor greater than three were not included together in the same model and our best judgement was used in determining which variable was more appropriate. GLMs were selected using Akaike's Information Criterion corrected for small sample size ($AIC_c$) [42,43] and the proportion of deviance explained was calculated using the *modEvA* package [49]. Models selected had an $AIC_c$ score at least 1.5 units smaller than the next closest model, with the exception of two instances where the top two models were within 0.4 units of one another. In these two instances, the model that explained more deviance was selected. To examine the relative importance of each predictor variable in the best fitted model, all possible

sub-model combinations of the final variable set were fitted and AIC$_c$ scores were calculated for each [42]. From this suite of models, the Akaike weights that included each variable were summed and plotted.

## Results

### Spatial distribution and abundance of adult shrimp

The spatial distribution of adult shrimp within the Pamlico Sound (1987–2019) varied on an annual basis and appeared to accurately reflect survey catch data (Table 2). The deviance explained by the GAMs varied by species and season but ranged from 46% to 74%. Fall and summer brown shrimp shared the same model formulation as did fall white shrimp and summer pink shrimp, while fall pink shrimp shared the same construction as for brown shrimp but used a higher *k* value (Table 2). Model output was used to project seasonal and annual maps of predicted shrimp distributions within the Pamlico Sound (e.g., brown shrimp in 2019, Fig 2). Much like mean survey CPUE and commercial catch data (S2 and S3 Figs), these estimates show a high degree of variability in annual and seasonal estimates of abundance but show that in recent years there have been increasing numbers of white shrimp in the fall, decreasing abundance of fall pink shrimp from a high period during the 1990s, and periodic high abundance of summer brown shrimp (Fig 3).

### Juvenile brown shrimp recruitment

Using 34 years of catch data from the P120 survey (1986–2019) and environmental data the best model to predict annual recruitment of juvenile brown shrimp explained 81% of the variability in the dataset (Table 3). This model used the June CPUE of juvenile brown shrimp and included *in situ* mean salinity and temperature, the interaction between north-south and east-west windstress during spawning and recruitment, NAO phase during the spawning and recruitment period, and an index of spawning adult biomass using commercial catch from the previous year (Fig 4a–4f). Using an index of relative importance, salinity, temperature, east-west windstress, and NAO phase were the most important predictors of recruitment (Fig 4g). All variables included in the best model had a positive correlation with the CPUE of juvenile brown shrimp except for east-west windstress (i.e., an offshore or west wind was associated with higher juvenile brown shrimp abundance). The significant interaction between east-west and north-south windstress demonstrated that a southwest-wind during the spawning and recruitment time period was associated with increased juvenile brown shrimp in the P120 survey (Fig 5a).

**Table 2. Parameters of chosen generalized additive models estimating the spatial distribution of adult brown (*Penaeus aztecus*), white (*P. setiferus*), and pink (*P. duorarum*) shrimp.**

| Response Variable | Deviance Explained | Model Structure | Tweedie Power Parameter | REML | Total Degrees of Freedom |
|---|---|---|---|---|---|
| Summer Brown Shrimp Log CPUE | 74.0% | ~s(Longitude, Latitude, k = 50) + Year + ti(Longitude, Latitude by = Year)) | 1.127 | 1047.608 | 162.86 |
| Fall Brown Shrimp Log CPUE | 46.0% | ~s(Longitude, Latitude, k = 50) + Year + ti(Longitude, Latitude by = Year)) | 1.113 | 1579.896 | 138.61 |
| Fall White Shrimp Log CPUE | 56.2% | ~s(Longitude, Latitude, k = 50)+s(YearNumeric, k = 30)+ti (Longitude, Latitude, YearNumeric, d = c(2,1), k = c(50,30))) | 1.123 | 1626.110 | 148.64 |
| Summer Pink Shrimp Log CPUE | 64.3% | ~s(Longitude, Latitude, k = 50)+s(YearNumeric, k = 30)+ti (Longitude, Latitude, YearNumeric, d = c(2,1), k = c(50,30))) | 1.091 | 1299.487 | 105.61 |
| Fall Pink Shrimp Log CPUE | 57.5% | s(Longitude, Latitude, k = 100) + Year + ti(Longitude, Latitude, by = Year)) | 1.111 | 1281.476 | 153.27 |

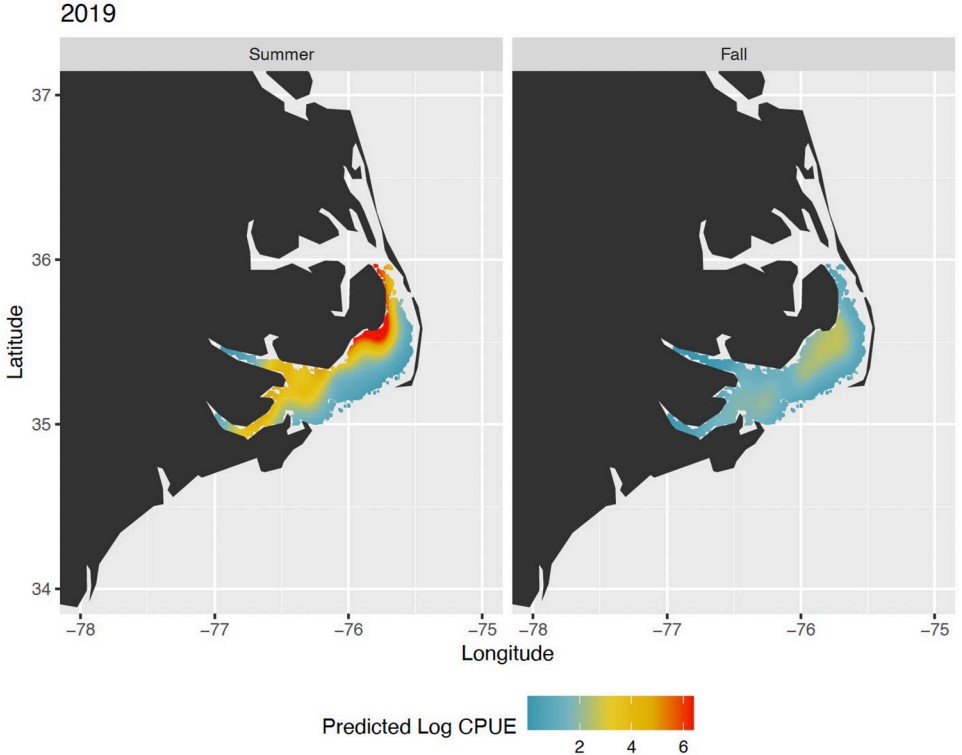

**Fig 2. The predicted spatial distribution of adult brown shrimp (*Penaeus aztecus*) in Pamlico Sound, North Carolina in the summer and fall of 2019. This year and species were selected as an example.** The colored region on the map shows the extent of the trawl survey area and the colors depict the predicted log catch per unit effort of adult brown shrimp using a generalized additive modeling approach. The spatial distribution of each species (brown, white, and pink) of shrimp was estimated for each year and season (1987–2019; summer and fall).

## Effect of climate and environmental variability on the abundance of adult shrimp

Estimated seasonal and annual abundance data from species-specific spatial GAMs were used in concert with environmental variables to examine the impact of climate on adult shrimp abundance. These models explained a high degree of the variability in the data (60–75%) with the exception of the model estimating fall pink shrimp abundance, which explained a more moderate 37% of the variability in the data (Table 3).

Adult summer brown shrimp abundance was predicted by the mean May CPUE from the P120 survey (estimate of nursery habitat recruitment), the north-south windstress during April, the *in situ* P195 mean summer temperature, and the SEAMAP CPUE from North Carolina waters during the previous fall (presumed spawning biomass; Fig 6a–6d). Relative variable importance demonstrated that the P120 May CPUE, April north-south windstress (a north wind is associated with higher abundance), and the *in situ* temperature were the most important variables within the model (Fig 6e). For adult fall brown shrimp abundance the best fitted model included spring temperature, an interaction between the east-west and north-south windstress during spawning and recruitment, the May CPUE from the P120 survey, and the NAO phase during the spawning and recruitment period (Fig 7a–7e). All the variables within the fall model had a relatively high variable importance (>0.5) indicating the importance of each of these predictors in explaining the variation in the fall abundance of adult brown

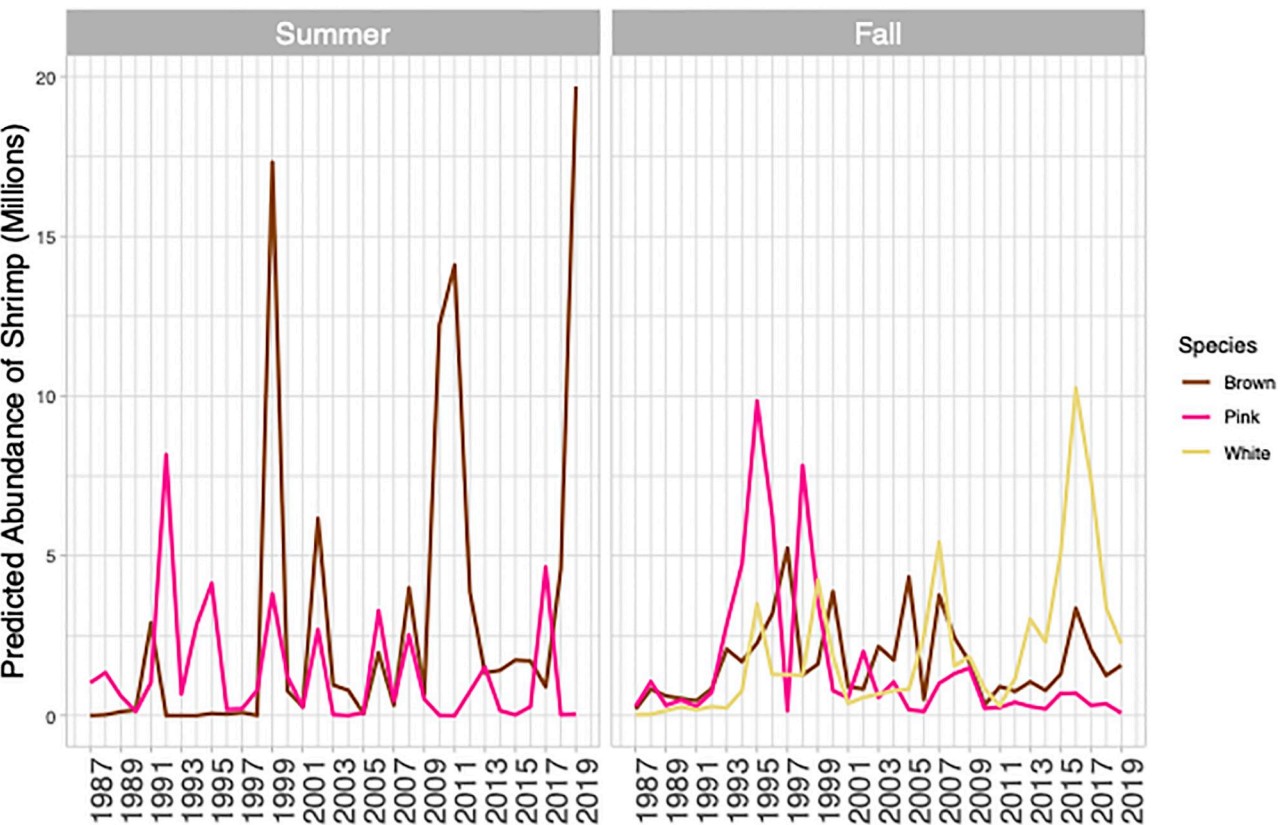

**Fig 3. Predicted abundance (millions) of adult brown (*Penaeus aztecus*), white (*P. setiferus*), and pink shrimp (*P. duorarum*) in Pamlico Sound, North Carolina from 1987–2019 for summer (June) and fall (September) using generalized additive spatial models.**

shrimp (Fig 7f). Similar to the model predicting juvenile brown shrimp abundance, a significant interaction between east-west and north-south windstress demonstrated a southwest wind during the spawning and recruitment period was associated with higher numbers of adult brown shrimp in the fall (Fig 5b). Respectively, the summer and fall models for adult brown shrimp predicted 66 and 65% of the variability in the data (Table 3).

The best model predicting the abundance of fall white shrimp explained 75% of the variation in the data using only the SEAMAP CPUE from North Carolina waters in the previous spring (presumed spawning biomass) and the December north-south windstress (i.e., a north wind was associated with higher numbers of adult white shrimp; Fig 8a and 8b). With only two variables, the relative variable importance was similar for both, but the SEAMAP CPUE was the most important for model performance (Fig 8c). Although winter temperature was not included in the best model for white shrimp abundance within the Pamlico Sound, the biomass of spawning white shrimp (spring SEAMAP CPUE) was significantly and positively correlated with winter temperature (linear regression, $p < 0.01$; estimate = 0.40; standard error = 0.14; t value = 2.96, $R^2$ = 0.21), suggesting a link between mild winters, more adult white shrimp in North Carolina waters, and subsequent higher recruitment in the Pamlico Sound.

The best model to explain age-one summer pink shrimp abundance included winter temperature, the lagged commercial catch index (presumed spawning biomass), east-west windstress from the previous spawning and recruitment period, and the P195 mean summer salinity

**Table 3. Variables included for best fit generalized linear models predicting the abundance of juvenile and adult brown (*Penaeus aztecus*), white (*P. setiferus*), and pink (*P. duorarum*) shrimp species.**

| Response Variable | Deviance Explained | Explanatory Variables | Parameter Estimates | Standard Error | $t$-value | $p$-value |
|---|---|---|---|---|---|---|
| Juvenile Brown Shrimp—June CPUE | 81% | *In situ* Salinity | 0.030 | 0.010 | 5.462 | <0.0001 |
| | | *In situ* Temperature | 0.109 | 0.020 | 6.913 | <0.0001 |
| | | Recruitment NAO | 0.052 | 0.020 | 2.115 | <0.05 |
| | | Recruitment $NAO^2$ | 0.028 | 0.010 | 1.891 | 0.07 |
| | | $\tau e$ | 88.360 | 44.100 | 2.004 | 0.06 |
| | | $\tau n$ | 73.400 | 23.390 | 3.138 | <0.01 |
| | | $\tau e \times \tau n$ | 34730.000 | 10690.000 | 3.250 | <0.01 |
| | | Lag Commercial Catch | 0.037 | 0.020 | 1.964 | 0.06 |
| Adult Brown Shrimp—Summer | 66% | P120 May Mean CPUE | 0.07511 | 0.01465 | 5.128 | <0.0001 |
| | | *In situ* Temperature | 0.03899 | 0.01792 | 2.176 | <0.05 |
| | | April $\tau n$ | 78.83638 | 36.76645 | 2.144 | <0.05 |
| | | Lag NC SEAMAP Fall CPUE | 0.04395 | 0.02519 | 1.745 | 0.09 |
| Adult Brown Shrimp—Fall | 65% | P120 May Mean CPUE | -0.020 | 0.010 | -3.229 | <0.01 |
| | | $\tau e$ | 30.560 | 16.250 | 1.880 | 0.07 |
| | | $\tau n$ | 25.080 | 8.340 | 3.006 | <0.01 |
| | | $\tau e \times \tau n$ | 10370.000 | 3424.000 | 3.027 | <0.01 |
| | | Spring Temperature | 0.230 | 0.070 | 3.156 | <0.01 |
| | | Spring $Temperature^2$ | -0.010 | 0.000 | -2.925 | <0.01 |
| | | Recruitment NAO | 0.020 | 0.010 | 2.130 | <0.05 |
| | | Recruitment $NAO^2$ | -0.010 | 0.010 | -2.078 | <0.05 |
| Adult White Shrimp—Fall | 75% | NC SEAMAP Spring CPUE | 0.059 | 0.007 | 8.364 | <0.0001 |
| | | Lag December $\tau n$ | 49.064 | 13.238 | 3.706 | <0.001 |
| Adult Pink Shrimp—Summer | 60% | Winter Temperature | 0.066 | 0.023 | 2.954 | <0.01 |
| | | Lag $\tau e$ | 76.952 | 28.585 | 2.692 | <0.05 |
| | | Lag Commercial Catch | 0.226 | 0.086 | 2.642 | <0.05 |
| | | *In situ* Salinity | 0.019 | 0.008 | 2.285 | <0.05 |
| Adult Pink Shrimp—Fall | 37% | $\tau e$ | 24.103 | 16.780 | 1.481 | 0.16 |
| | | $\tau n$ | -38.164 | 16.370 | -2.331 | <0.05 |
| | | Lag NC SEAMAP Fall CPUE | 0.007 | 0.020 | 0.368 | 0.72 |
| | | Recruitment River Volume | 0.031 | 0.018 | 1.741 | 0.09 |

*P120* is the Program 120 survey that samples juvenile shrimp within tidal creeks in Pamlico Sound. *In situ Temperature* and *Salinity* refer to the temperatures and salinity measured by either the juvenile or adult shrimp survey, whereas *Winter Temperature* and *Spring Temperature* refer to water temperatures measured by NOAA buoys during referenced season. $\tau e$ is the East-West wind stress and $\tau n$ is the North-South wind stress. *Commercial Catch* refers to the standardized catch of shrimp (weight in pounds) within Pamlico Sound. *NC SEAMAP CPUE* refers to the log catch per unit effort of shrimp on the North Carolina shelf caught during the Southeast Area Monitoring and Assessment Program trawl survey. *Recruitment* is the species-specific time period during which post-larval shrimp recruit to Pamlico Sound.

in the Pamlico Sound (Fig 9a–9d). This model explained 60% of the variation in the data (Table 3). Winter water temperature was the most important predictor of summer pink shrimp abundance but each of the variables in the model had a relative importance (> 0.5) illustrating their role in predicting summer pink shrimp abundance (Fig 9e). The best model to explain fall abundance of pink shrimp explained 37% of the variability in the data (Table 3). This model included north-south windstress during spawning and recruitment (i.e., a south wind is associated with higher abundance of fall adult pink shrimp), river flow during spawning and recruitment, east-west windstress during spawning and recruitment (i.e., an onshore

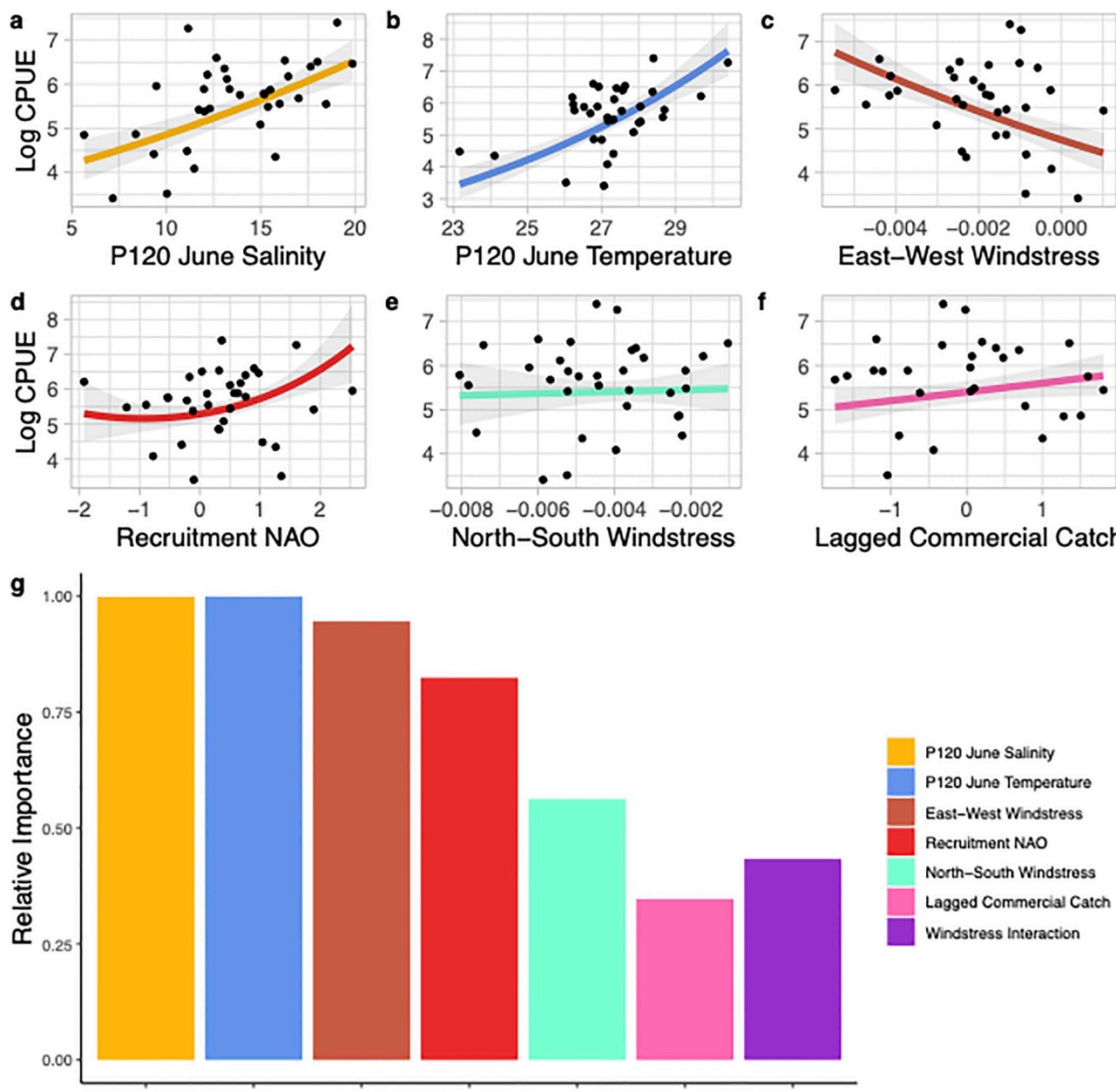

**Fig 4. Environmental and spawning stock variables that were included in the best fit generalized linear model predicting the log June CPUE of juvenile brown shrimp (*Penaeus aztecus*) caught in the P120 survey in Pamlico Sound, North Carolina from 1986–2019.** Panels a-f show the relationship between each variable included in the model and the log catch per unit effort (log CPUE; solid line) with the 95% confidence intervals shown in shaded grey. Panel g shows the relative importance of each variable included in the best model. It should be noted that panels c and e show directional windstress components but see Fig 5a for their interaction.

or east wind is associated with higher abundance of fall adult pink shrimp), and the SEAMAP CPUE from North Carolina waters in the previous fall (presumed spawning biomass; Fig 10a–10d). The most important variables within the model were north-south windstress, river volume, and east-west windstress (Fig 10e).

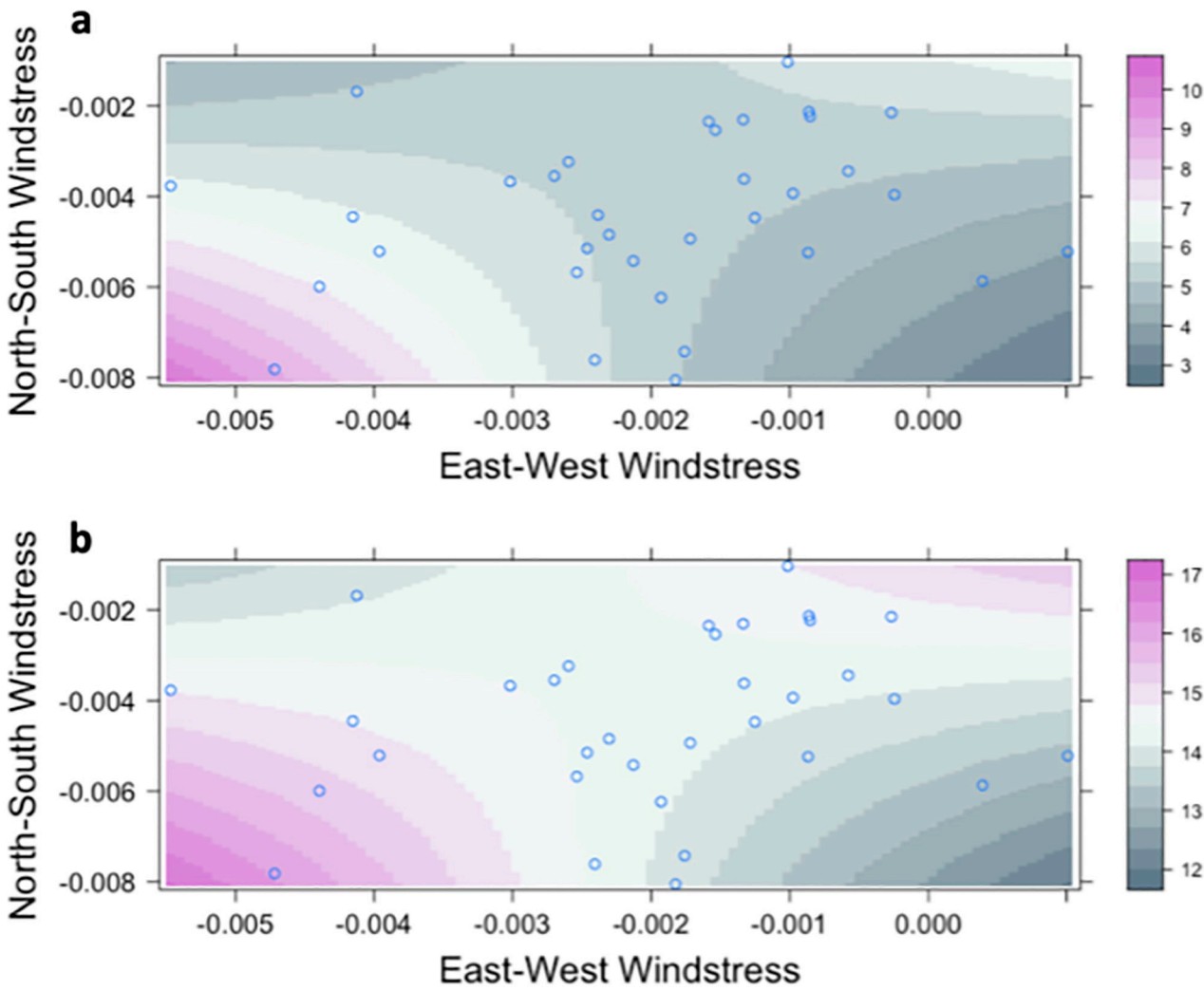

**Fig 5. East-west and north-south windstress interactions that were included in the best generalized linear models predicting the a) abundance of spring juvenile and b) fall adult brown shrimp (*Penaeus aztecus*) in Pamlico Sound, North Carolina.** Colored contours show the log and predicted log catch per unit effort (CPUE) for juvenile and adult brown shrimp, respectively, and the blue circles show the measured wind stress values. More negative values of the east-west and north-south wind stress represent wind from the west and south, respectively.

## Discussion

Penaeid shrimp species are of enormous ecological, cultural, and economical importance to North Carolina. As an estuarine-dependent species with an annual life cycle, the relationship between penaeid shrimp abundance and environmental conditions is tightly linked and adult shrimp abundance may serve as an important bioclimatic indicator that will prove increasingly important to evaluate the impact of climate change on coastal ecosystems. The findings from our analyses indicate that the successful recruitment of juvenile brown shrimp and abundance of adult penaeid shrimp in North Carolina is highly dependent on favorable environmental conditions. Additionally, nearly all our models included a measure of spawning stock biomass from the previous season, illustrating the important connection between abundant spawners and a strong recruitment class. In particular, all three species responded strongly to

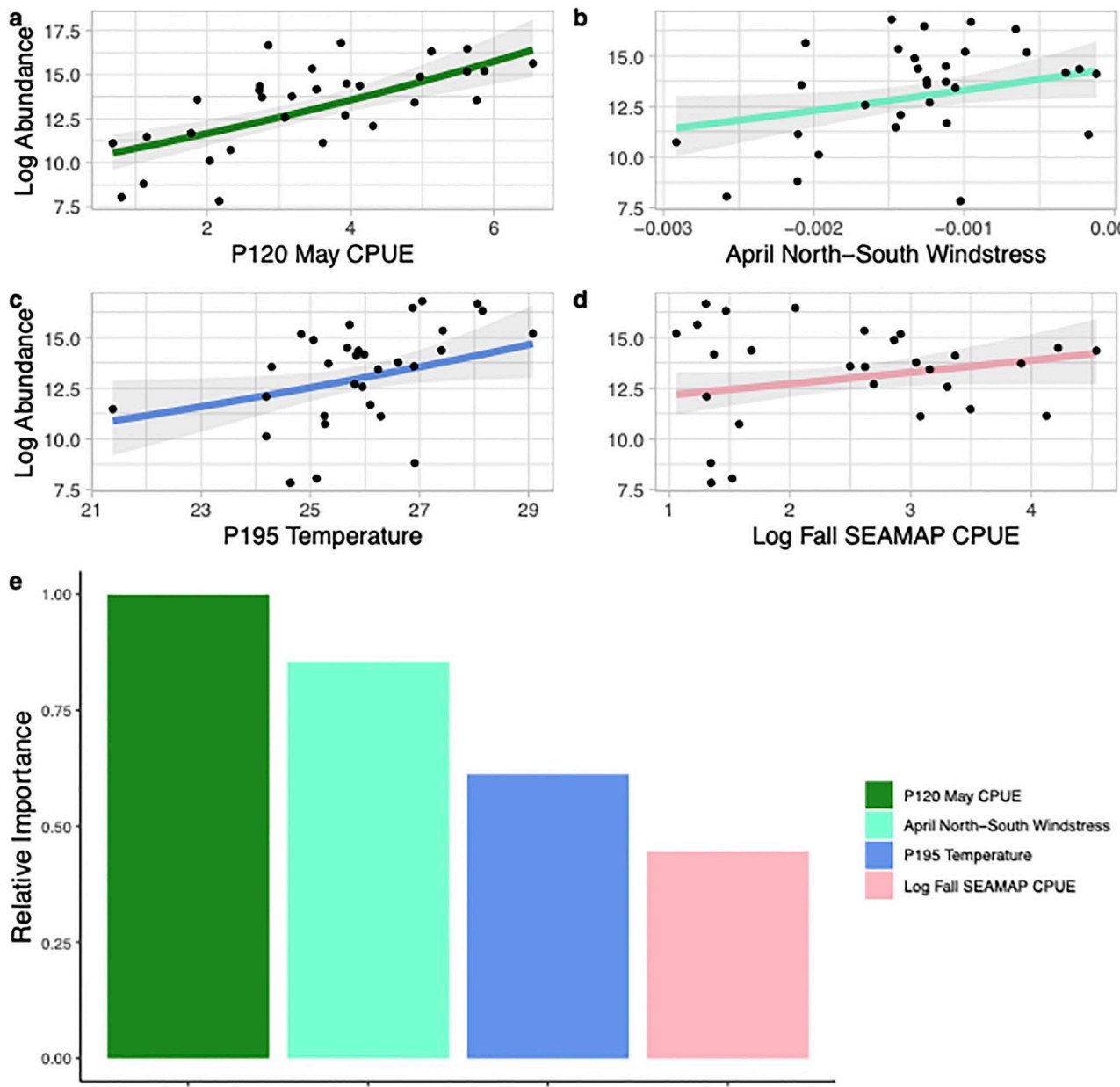

**Fig 6. Environmental and spawning stock variables that were included in the best generalized linear model predicting the estimated log summer abundance of adult brown shrimp (*Penaeus aztecus*) in Pamlico Sound, North Carolina from 1987–2019.** Panels a-d show the relationship between each variable included in the model and the estimated log abundance (solid line) with the 95% confidence intervals shown in shaded grey. Panel e shows the relative importance of each variable included in the best model.

temperature, salinity, windstress, and the presumed spawning biomass while the NAO index was particularly important for predicting the abundance of brown shrimp.

Of each of our models, our study found the highest degree of variability (81%) explained by our juvenile brown shrimp model (Table 3). Given that at the juvenile life stage there has been less time for factors outside of model variables such as predation pressure and food availability to influence abundance, this high degree of variance explained makes sense. Additionally, given that catches of juvenile brown shrimp at the P120 fixed stations are inconsistent from

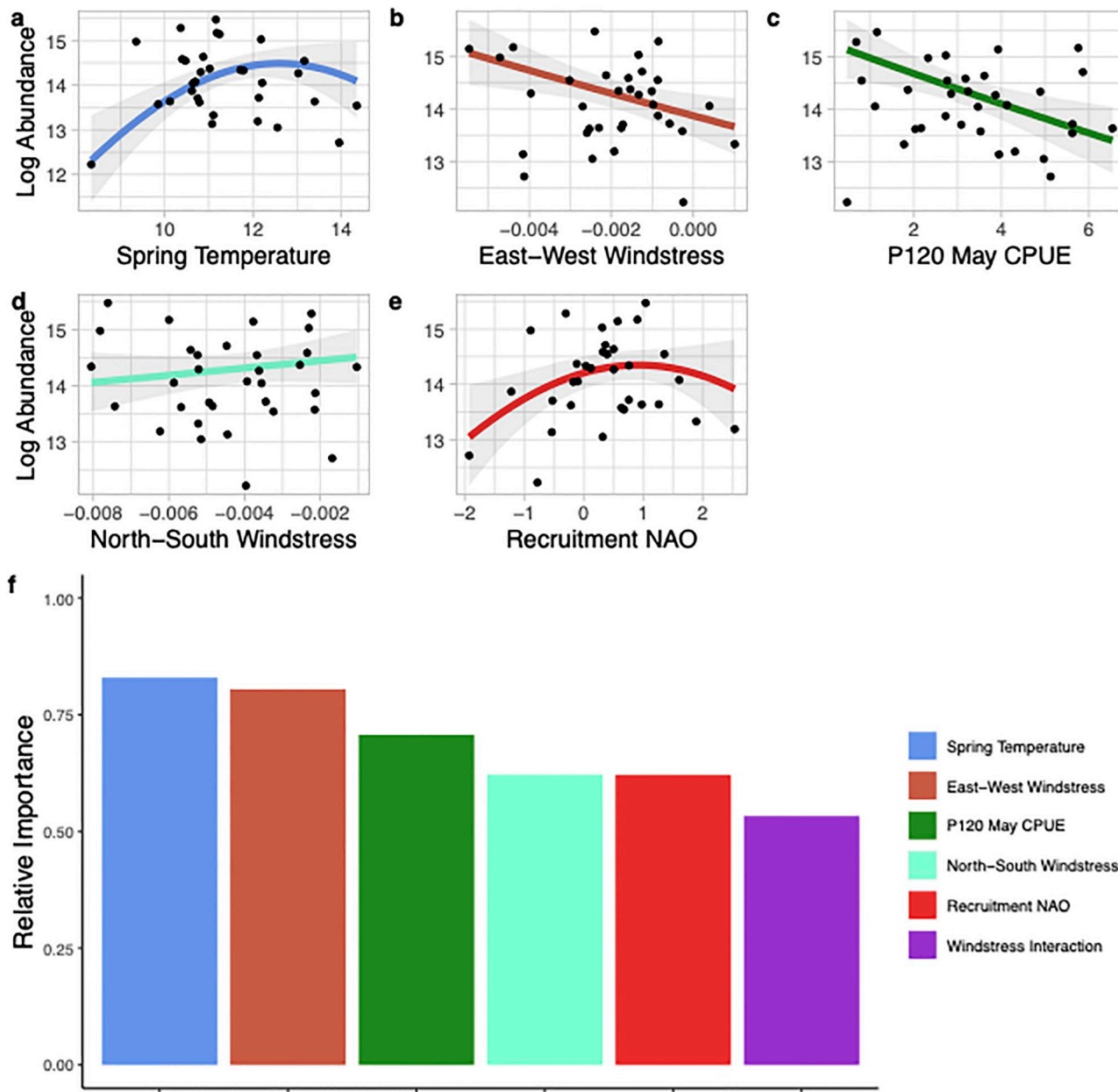

**Fig 7. Environmental and spawning stock variables that were included in the best generalized linear model predicting the estimated log fall abundance of adult brown shrimp (*Penaeus aztecus*) in Pamlico Sound, North Carolina from 1987–2019.** Panels a-e show the relationship between each variable included in the model and the estimated log abundance (solid line) with the 95% confidence intervals shown in shaded grey. Panel f shows the relative importance of each variable included in the best model. It should be noted that panels b and d show directional windstress components but see Fig 5b for their interaction.

year to year [50], this illustrates the large role of environmental variability and spawning stock biomass in successful recruitment. Successful recruitment of juveniles to a population frequently plays a key role in population dynamics; however, in the case of short-lived species, the recruiting year class contributes a significant share of the landings and a juvenile recruitment index can be useful for determining total allowable catches [51]. In fact, the juvenile index of abundance was one of the most important variables in predicting the summer and fall

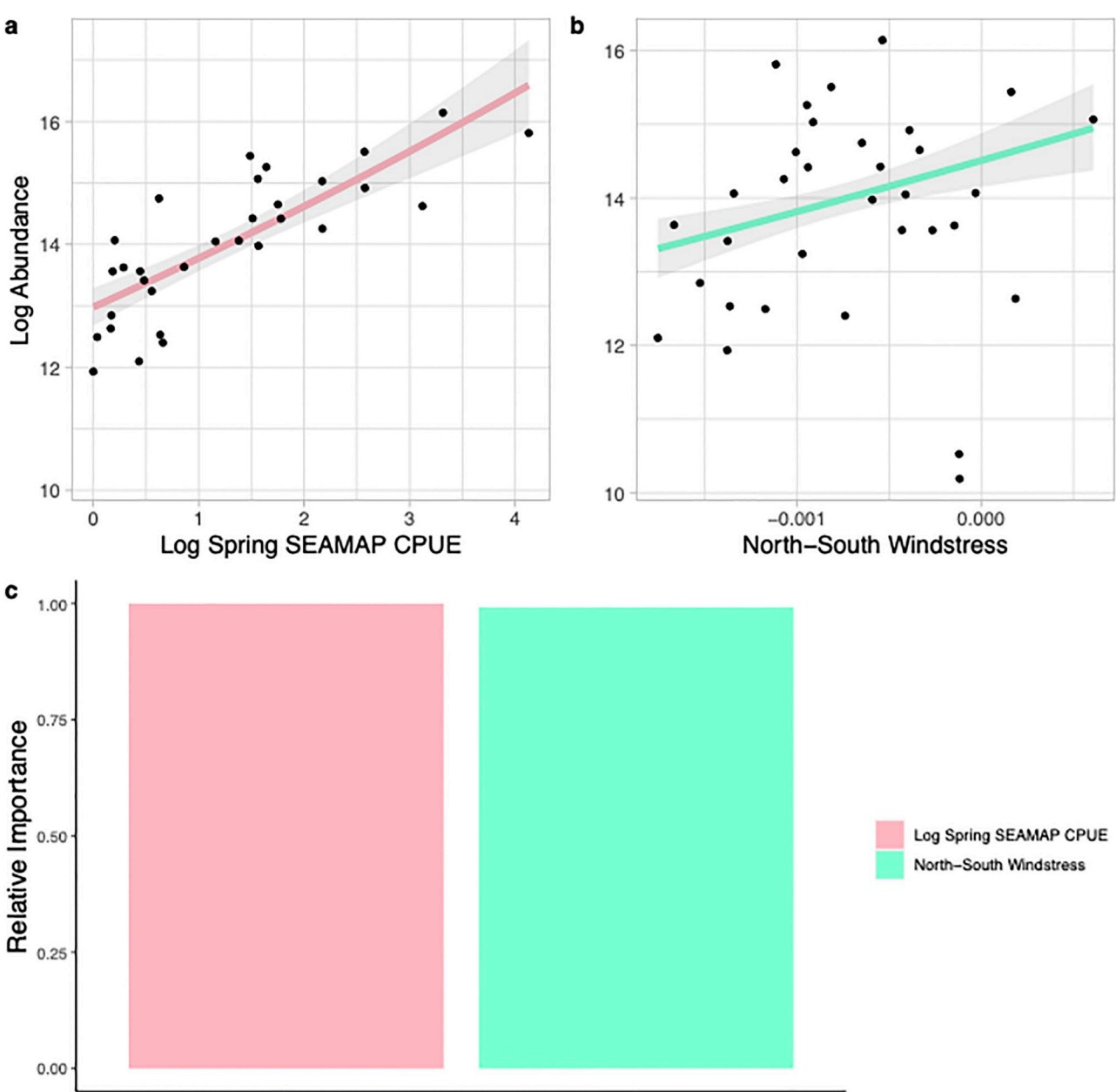

**Fig 8. Environmental and spawning stock variables that were included in the best generalized linear model predicting the estimated log fall abundance of adult white shrimp (*Penaeus setiferus*) in Pamlico Sound, North Carolina from 1987–2019.** Panels a-b show the relationship between each variable included in the model and estimated log abundance (solid line) with the 95% confidence intervals shown in shaded grey. Panel c shows the relative importance of each variable included in the best model.

abundance of adult brown shrimp in the Pamlico Sound. Although we were not able to examine the relationship between juvenile white or pink shrimp and adult biomass due to the timing of the P120 survey, we would expect that those relationships would be similarly as strong as has been seen in other systems [52].

Although the deviance explained by our juvenile brown shrimp abundance model was higher than for our adult abundance models, adult models still explained a large proportion of variation in the data. Previous studies of shrimp in the Gulf of Mexico found the influence of

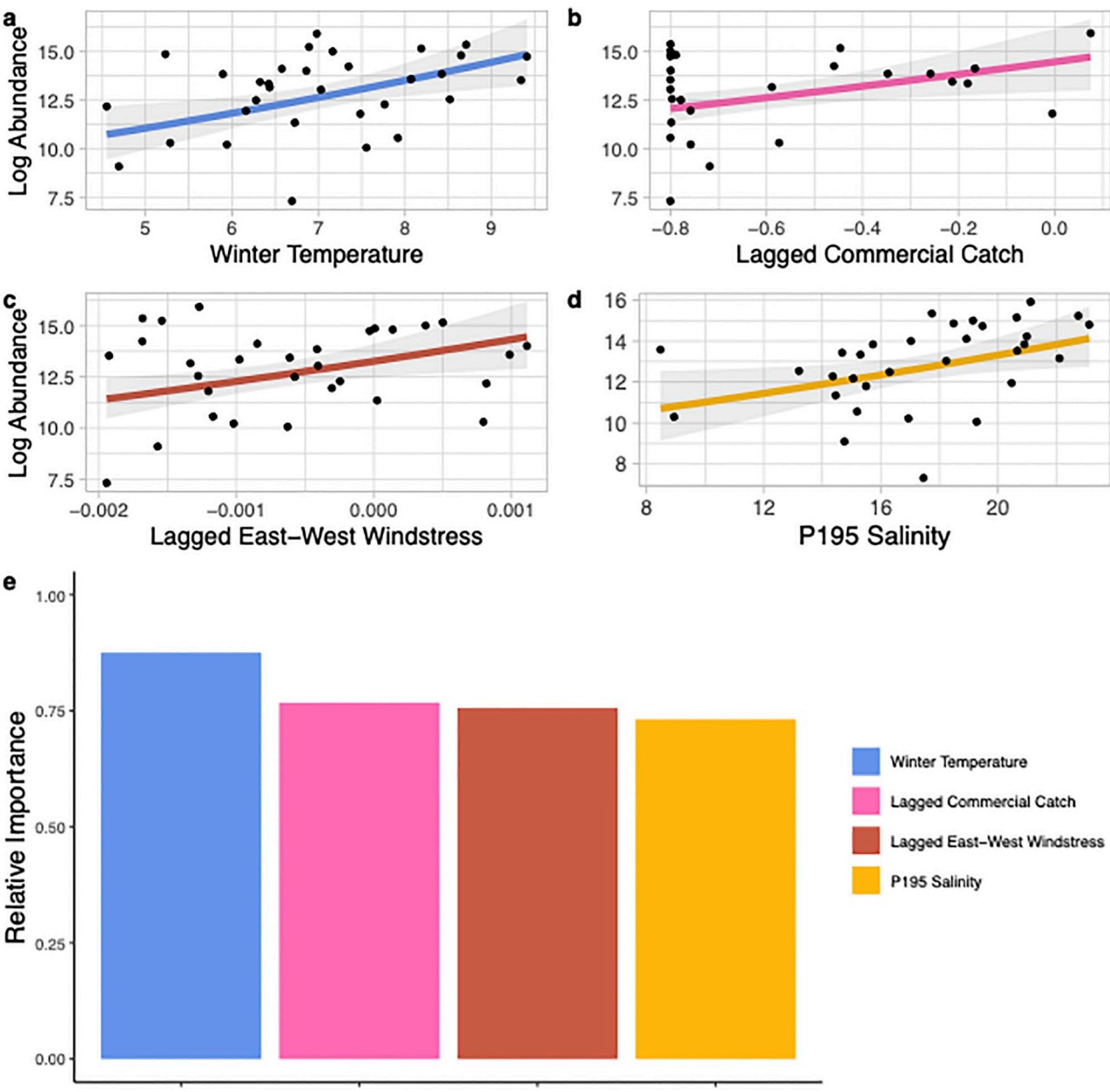

**Fig 9. Environmental and spawning stock variables that were included in the best generalized linear model predicting the estimated log summer abundance of adult pink shrimp (*Penaeus duorarum*) in Pamlico Sound, North Carolina from 1987–2019.** Panels a-d show the relationship between each variable included in the model and estimated log abundance (solid line) with the 95% confidence intervals shown in shaded grey. Panel e shows the relative importance of each variable included in the best model.

environmental factors had largely eroded for adult shrimp [52], this was not the case in our study and, with the exception of fall pink shrimp, models for adult shrimp explained at least 60% of the variation in the data. The Pamlico Sound is a large lagoonal estuary where a large area of shrimp nursery habitat is accessed through few oceanic inlets. Therefore, the adult shrimp in this system may represent a more distinct cohort compared to other systems like the Gulf of Mexico, where more heterogeneous nursery habitats may contribute adult shrimp.

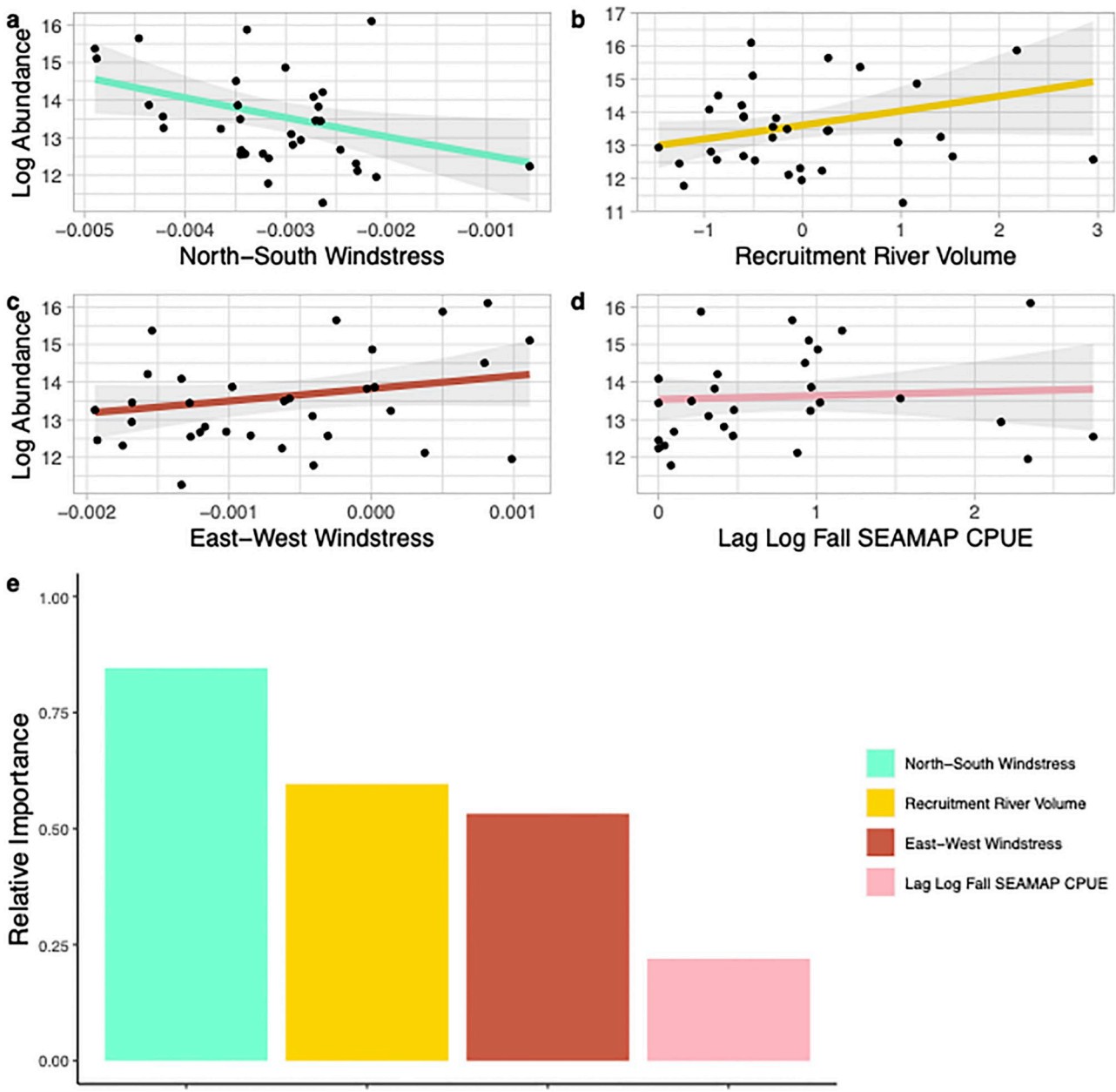

**Fig 10. Environmental and spawning stock variables that were included in the best generalized linear model predicting the estimated log fall abundance of adult pink shrimp (*Penaeus duorarum*) in Pamlico Sound, North Carolina from 1987–2019.** Panels a-d show the relationship between each variable included in the model and the estimated log abundance (solid line) with the 95% confidence intervals shown in shaded grey. Panel e shows the relative importance of each variable included in the best model.

Intense commercial trawling for shrimp occurs in the Pamlico Sound during summer and fall [33], which has the potential to erode model predictive performance for estimates based on the September survey; however, fall models for brown and white shrimp still had 65% or greater deviance explained. Our model estimating the abundance of fall pink shrimp explained 37% of the variation in the data. The relatively weaker performance of that model together with the decline in abundance of pink shrimp since the 1990s suggests a factor outside the bounds of our model is likely influencing the success of pink shrimp in the Pamlico Sound. The increase

in white shrimp in the Pamlico Sound that has occurred over approximately the same period, could suggest interspecific competition is contributing to the decline of pink shrimp. However, it should be noted that many other factors not included in our models such as loss of seagrass [53], predation pressure, or shifts in distribution outside of Pamlico Sound could also explain declining numbers of pink shrimp.

The NAO index represents a climate pattern that encompasses several environmental conditions including windstress, precipitation (and therefore salinity in estuarine ecosystems), storm intensity, circulation patterns, and oceanic heat transport [54,55]. For the past several decades, the NAO index has been primarily in a positive phase, which is associated with warmer and less stormy conditions on the U.S. east coast [54,55]. Intermediate to positive NAO phases were found to be beneficial for both juvenile and adult fall brown shrimp. The NAO index has previously been shown to be associated with shifts in marine species assemblages [56] and has been associated with changes in community composition and residency patterns in estuarine ecosystems [9,57]. Our results add to the body of work demonstrating the role of the NAO, and broader climate patterns, on estuarine-dependent species.

For estuarine-dependent species such as penaeid shrimp, variations in salinity throughout ontogeny can represent a highly dynamic environmental variable. High salinity had a positive effect on abundance for juvenile brown and one-year old summer pink shrimp while fall pink shrimp were more abundant with increased river volume (lower salinity). However, the relationship between fall pink shrimp and river volume was weaker than the relationship between salinity and summer pink shrimp. As mentioned previously, summer pink shrimp are age-one and have overwintered in the Pamlico Sound [45], while pink shrimp caught in the fall have recently recruited and it is not unexpected that they would show ontogenetic differences in their response to environmental variables.

Windstress during spawning and recruitment periods was an important variable for predicting the abundance of brown and pink shrimp. Windstress can impact larval transport as well as the distribution and concentration of plankton [58], which may directly impact the abundance of shrimp in the Pamlico Sound. Wind direction and speed have previously been shown to influence the recruitment and success of shrimp in the southeast US and Gulf of Mexico [59–61] as well as other estuarine species in North Carolina [62,63].

In the coastal waters within the Middle and and South Atlantic Bight, moderate upwelling and offshore Ekman flow of surface waters ocurr when south and southwest winds predominate, while winds originating in the north and northeast cause onshore Ekman flow [59,64]. Therefore, brown shrimp tended to be abundant during periods of upwelling and pink shrimp were more abundant during periods of downwelling. While predominant offshore winds might seem to pose a challenge for the successful recruitment of larval and postlarval brown shrimp, position in the water column is also an important factor when considering wind-driven transport [59] and may indicate behavioral differences between the species. A deeper position in the water column is favorable for onshore transport during periods of offshore windstress while a more shallow orientation would be beneficial during onshore windstress. Brown shrimp are known to spawn in deeper waters (>64 m), while pink shrimp have been found to spawn in more shallow areas (<40 m) [65,66], indicating that spawning depth and larval and postlarval behavior influence recruitment success. These results suggest that larval pink shrimp are associated with the upper layers of the water column while larval brown shrimp remain deeper in the water column in order to recruit successfully.

Although southwest windstress was a key variable for predicting the June juvenile and September adult brown shrimp abundance, north windstress, specifically in April, when many brown shrimp are recruiting to the Pamlico Sound, was associated with high summer abundance. Brown shrimp spawning and recruitment in North Carolina takes place over a fairly

long period from January to April (Table 1) [32] and this apparent contrast in windstress directionalty between brown shrimp across seasons may suggest that onshore Ekman flow is particularly beneficial to brown shrimp at key ontogenetic stages and that different environmental conditions contribute to the success of different pulses of recruits within a year class. Finally, since brown shrimp are the earliest of the three species to recruit to the Pamlico Sound, the productivity associated with upwelling and the southwest windstress may be particularly important for their success. Overall, pink shrimp abundance showed an opposite response to windstress directionality than brown shrimp and this may partially explain why our model predictions suggest that high abundance of pink shrimp does not typically co-ocurr with high abundance of brown shrimp.

Fall white shrimp abundance was highly influenced by north windstress during the previous December, and this pattern was the only instance where a period of windstress at a time other than the spawning and recruitment period was important for predicting abundance. Previous studies have identified windstress as an important variable in determining abundance of estuarine species in the period immediately preceeding recruitment. For example, in South Carolina's Ashepoo-Combahee-Edisto (ACE) basin, northwest winds over the previous seven days were associated with low numbers of juvenile white shrimp while windstress two days prior to recruitment was an important factor in predicting blue crab (*Callinectes sapidus*) abundance in North Carolina [61,62]. However, in the present study December north windstress did not immediately precede recruitment and was not significantly correlated with winter water temperature; therefore we expect it may be related to an additional variable outside the bounds of our model that may influence the location of pre-spawning adults on the continental shelf, and therefore indirectly influence the abundance of white shrimp larvae that recruit to the Pamlico Sound.

For all three species of shrimp, abundance was positively correlated with temperature. Although for brown shrimp fall abundance, intermediate spring temperatures (~11–13 ˚C) were predicted to have the highest abundances (Fig 7a). This relationship could suggest a negative impact of temperature above a particular physiological tolerance; however, we feel it is more likely related to phenology. At the greatest spring temperatures, growth rates of juvenile brown shrimp were likely higher [67], indicating that by the fall P195 survey in September, many brown shrimp would have already reached maturity and left the Pamlico Sound. This is further supported by the negative relationship between May juvenile brown shrimp CPUE and fall brown shrimp abundance. The negative correlation between increased early recruitment and decreased fall abundance suggests that when brown shrimp recruit in high numbers to the Pamlico Sound in May they typically grow quickly and leave the sound before the September P195 survey occurs. Or, alternatively, that density dependent mortality or heightened interspecific competition occurs and results in decreased fall abundance. Overall, the positive relationship with species abundance and temperature suggests the potential for all three species to shift their distribution northward as ocean temperatures continue to increase. In fact, the potential for northward range expansion of brown shrimp on the US east coast has been documented back to 1938 where large numbers were caught in Great South Bay, New York following a particularly warm fall [68]. White shrimp have also been shown to move north and increase population biomass on the southeast continental shelf in response to mild winters [29,69]. In recent years white shrimp have been in greater abundance in northern regions prompting the Virginia Marine Resources Commission to establish a shrimp fishery in Virginia waters for the first time in 2019 [70].

Our analyses showed that cold winter water temperatures can also negatively affect both white and pink shrimp and these species are known to be sensitive to lower lethal limits in this region [29,46,61,69]. In South Carolina's ACE basin, winter temperatures below 11˚C had a

significant negative impact on the abundance of adult white shrimp [61] and the effect of temperature on this species is so well established that the South Carolina white shrimp fishery is closed when the temperature is lower than 9˚C for seven consecutive days as a measure to protect the spawning stock [71,72]. For pink shrimp, the timing of their recruitment to the Pamlico Sound (June to August; Table 1) means that most do not reach maturity in time to move offshore before winter, and instead overwinter within the Pamlico Sound [32,45]. Therefore, winter temperatures are likely to have a strong impact on their survival [46,73]. Our findings corroborate previous work showing water temperature from the previous winter is the most important factor determining the abundance of summer pink shrimp in North Carolina [46] and a driving factor in the abundance of white shrimp throughout the southeastern United States [61]. Brown shrimp have previously been hypothesized to be sensitive to colder winter temperatures based on the timing of their reproduction and postlarval recruitment [29], although in the present analyses spring or summer temperature was selected over winter temperature in model evaluation suggesting that brown shrimp are relatively robust to cold conditions [74].

The abundance and distribution of adult shrimp, i.e., the presumed spawning stock biomass, was also important in predicting the abundance of all three penaeid species, suggesting that in contrast to many marine species [28], the relationship between spawners and recruits can be significant and linear in annual penaeid species. The relationship between spawning biomass and shrimp in the Pamlico Sound was strongest for white shrimp where the spring CPUE of white shrimp captured in the SEAMAP survey on the North Carolina shelf was the most important variable explaining the estimated abundance of white shrimp in the Fall in the Pamlico Sound (Fig 8c). This same relationship has been observed in the Chesapeake Bay, where the SEAMAP CPUE of white shrimp is positively correlated with the CPUE for Virginia Institute of Marine Science (VIMS) trawl survey [70] and South Carolina's ACE basin [61], suggesting this relationship is not exclusive to North Carolina or local conditions. Similarly, researchers in the Chesapeake Bay found the relationship between indices of adult abundance and subsequent juvenile recruitment was strongest for white shrimp relative to brown and pink shrimp [70]. In the present study we found the spring spawning biomass of white shrimp was highly correlated with winter water temperature, suggesting that although the spawner-recruit relationship was dominant for white shrimp, temperature was an underlying factor in that relationship. Adult white shrimp are thought to migrate southward during winter. Our results along with Tuckey et al. (2021) and Morley et al. (2017), suggest that after mild winters the northern distribution of white shrimp overlaps strongly with North Carolina and Virginia, and when spawning occurs there is a greater supply of postlarvae to these large northern estuaries (i.e., the Pamlico Sound and Chesapeake Bay). Thus, these studies provide an example of how climate change, acting on a specific life stage and season, can lead to an emergent population level response across life stages.

The ultimate goal of understanding the relationship between population abundance and predictor variables is to have the potential to forecast changes in stock productivity [30]. Although a true forecast is beyond the scope of the present work, the importance of a recruitment index in predicting the abundance of summer and fall brown shrimp abundance may be useful to North Carolina fishery managers and fishermen in anticipating shrimp landings. Additionally, if effort in the fishery is influenced by shrimp abundance, then fishery managers might use predictions of shrimp abundance to anticipate other impacts of the fishery such as trawl bycatch. Overall, our models explained a high degree of data variability and demonstrate both environmental variables, particularly temperature and windstress, and spawning stock biomass are important in predicting the abundance of brown, white, and pink shrimp in North Carolina. In particular, our finding that the distribution and abundance of spawning

stock biomass from the previous year class is positively correlated with abundance may provide some guidance for fishery managers.

As an important prey species for many estuarine and oceanic predators [23–25], understanding variability in penaeid shrimp populations may aid efforts to use ecosystem-based fishery management [75]. Critically, as climate change continues to alter ocean and estuarine ecosystems it remains essential to understand the relationship between marine species and their environment in order to successfully implement adaptive management strategies [76]. Our analyses suggest penaeid shrimp abundance and population-level movement may be a particularly useful indicator of bioclimatic variability and may augment our understanding of complex population-level shifts in estuarine-dependent organisms.

## Supporting information

**S1 Fig. Length frequency distributions for brown (*Penaeus aztecus*), pink (*P. duorarum*), and white (*P. setiferus*) shrimp caught in the North Carolina Division of Marine Fisheries (NCDMF) P120 spring tidal creeks trawl survey (TC Survey-Spring) from 1986 to 2019 and the NCDMF P195 Pamlico Sound June (PS Survey-Summer) and September (PS Survey-Fall) trawl survey from 1987 to 2019.** This figure represents length data from 94,952, 13,775, and 15,732 brown shrimp; 1,815, 10,740, and 11,025 pink shrimp; and 2,662, 607, and 17,300 white shrimp, for the spring P120, summer P195, and fall P195 trawl surveys, respectively.
(TIFF)

**S2 Fig. Mean catch per unit of effort (CPUE) from the North Carolina Division of Marine Fisheries P195 trawl survey from 1987 to 2019 for brown (*Penaeus aztecus*), pink (*P. duorarum*), and white (*P. setiferus*) shrimp in the summer (June) and fall (September) survey that occurs in the Pamlico Sound, NC.**
(TIFF)

**S3 Fig. Commercial brown (*Penaeus aztecus*), pink (*P. duorarum*), and white (*P. setiferus*) penaeid shrimp landings from 1987–2019 within Pamlico Sound, North Carolina.**
(TIFF)

## Acknowledgments

We thank the North Carolina Division of Marine Fisheries for maintaining the P120 and P195 surveys and collecting the data used in this study.

## Author Contributions

**Conceptualization:** Lela S. Schlenker, Chris Stewart, Jason Rock, Nadine Heck, James W. Morley.

**Data curation:** Lela S. Schlenker, Chris Stewart, Jason Rock, James W. Morley.

**Formal analysis:** Lela S. Schlenker.

**Funding acquisition:** Nadine Heck, James W. Morley.

**Investigation:** Lela S. Schlenker, James W. Morley.

**Methodology:** Lela S. Schlenker, Chris Stewart, Jason Rock, James W. Morley.

**Project administration:** James W. Morley.

**Resources:** James W. Morley.

**Supervision:** Nadine Heck, James W. Morley.

**Visualization:** Lela S. Schlenker.

**Writing – original draft:** Lela S. Schlenker, James W. Morley.

**Writing – review & editing:** Chris Stewart, Jason Rock, Nadine Heck.

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
