## [Decision Letter · Decision Letter 0]

8 Dec 2022

PONE-D-22-30838Environmental and climate variability drive population size of annual penaeid shrimp in a large lagoonal estuaryPLOS ONE

Dear Dr. Schlenker,

Thank you for submitting your manuscript to PLOS ONE. After careful consideration, we feel that it has merit but does not fully meet PLOS ONE’s publication criteria as it currently stands. Therefore, we invite you to submit a revised version of the manuscript that addresses the points raised during the review process.

We have got evaluations from three well qualified experts. In particular Rev1 has done a thorough job looking critically into your statistical analyses and modelling and providing good advise. 

I and at least one new or one of the original reviewers will evaluate your revised manuscript. 

We look forward to receiving your revised manuscript.

Kind regards,

Geir Ottersen

Academic Editor

PLOS ONE

"Funding for this research was provided in a grant to J.W.M. and N.H. from North Carolina Sea Grant."

"This research was supported by a grant from North Carolina Sea Grant (NCSG-RM-02) to JWM and NH. The funders had no role in study design, data collection and analysis, decision to publish, or preparation of the manuscript."

4. We note that Figures 1 and 2 in your submission contain [map/satellite] images which may be copyrighted. All PLOS content is published under the Creative Commons Attribution License (CC BY 4.0), which means that the manuscript, images, and Supporting Information files will be freely available online, and any third party is permitted to access, download, copy, distribute, and use these materials in any way, even commercially, with proper attribution. For these reasons, we cannot publish previously copyrighted maps or satellite images created using proprietary data, such as Google software (Google Maps, Street View, and Earth). For more information, see our copyright guidelines: http://journals.plos.org/plosone/s/licenses-and-copyright.

a. You may seek permission from the original copyright holder of Figures 1 and 2 to publish the content specifically under the CC BY 4.0 license.  

Reviewers' comments:

Reviewer's Responses to Questions

**Comments to the Author**

1. Is the manuscript technically sound, and do the data support the conclusions?

Reviewer #1: Partly

Reviewer #2: Yes

Reviewer #3: Yes

2. Has the statistical analysis been performed appropriately and rigorously? 

Reviewer #1: No

Reviewer #2: Yes

Reviewer #3: I Don't Know

3. Have the authors made all data underlying the findings in their manuscript fully available?

Reviewer #1: Yes

Reviewer #2: Yes

Reviewer #3: Yes

4. Is the manuscript presented in an intelligible fashion and written in standard English?

Reviewer #1: Yes

Reviewer #2: Yes

Reviewer #3: Yes

5. Review Comments to the Author

Reviewer #1: Please find my review of manuscript ID PONE-D-22-30838 entitled “Environmental and climate variability drive population size…” by Schlenker et al. This investigation explored the relationship between various environmental factors and their influence on the abundance of adult brown, pink, and white shrimp and juvenile brown shrimp in Pamlico Sound. The significant relationship between previous spawning stock sizes and the abundance realized in the fisheries independent trawl surveys lends credence the widely-held conclusion that the spawner-recruit relationship for short-lived species is directly related. They also found interesting associations between environmental factors and abundance of penaeid shrimp. The discussion section of this manuscript was extremely thorough, provided many thoughtful links to the literature, and posited compelling hypotheses. This study provides valuable information on potential drivers of abundance and recruitment of these species. However, I am very concerned about the rigor and validity of the statistical methods employed in this analysis.

My most pressing statistical concern, that needs to be addressed to proceed with publication, is regarding the model of juvenile brown shrimp abundance. If I am properly understanding the authors’ methods, they fit GLMs to the average annual CPUE from May, June, and combined May and June, then compared these three models to one another to determine which GLM had the largest deviance explained. The response data used in each of these three models is different, and therefore, they cannot be compared to one another. If the response data are not shared amongst a set of models, then it is not statistically sound to compare these models and choose the “best” fit. This also means that the conclusions are overstated and misleading. The GLM does not provide information on the relationship between the environmental covariates and brown shrimp recruitment, as suggested by the authors, but instead provides information on the relationship between the environmental factors and CPUE of juvenile brown shrimp in June only. These same environmental factors may not have displayed a relationship with the CPUE of juvenile brown shrimp in May, or the CPUE for the spring. As a reader, it is impossible to know since the results of these models were not reported. I would suggest using the average annual CPUE for the spring (i.e., May and June) if the authors intend to draw conclusions on the overall relationship between the environmental factors and recruitment of juvenile brown shrimp. If I am misunderstanding the methods employed for this analysis, then the methods and results need to be rewritten to clarify that proper statistical procedures were followed.

An additional important revision is regarding the conclusions drawn between windstress and juvenile brown shrimp CPUE in June and adult brown shrimp abundance in the fall. Both of these models included significant interactions between east-west and north-south windstresses. Because of the significant interaction, the relationship between abundance and one windstress cannot be described without including the other windstress. That is, the influence of north-south windstress (as an example) on abundance is dependent upon the east-west windstress. To continue the example, figure 5 demonstrates that lower values of north-south windstress have a positive relationship with juvenile brown shrimp abundance in June, but only when east-west windstress is also low. The authors describe the relationship between abundance and each of the windstresses individually in the text, and display the individual relationships in figures (4 and 7), without the inclusion of the other directional windstress when doing so. The interaction also needs to be considered in the discussion section of the manuscript.

The spatiotemporal modeling section is interesting, but the benefit to using a spatiotemporal model is never described (e.g., is there spatial and/or temporal autocorrelation in the data?). Additionally, there was no spatial modeling of the juvenile brown shrimp, only adults, and it is not explained as to why the juvenile brown shrimp model doesn’t follow the same methods as the adult models. Further, a description as to why the output of the spatial models was summed for use in the GLMs (i.e., so the response variable is on the annual scale for evaluation) would be useful for the reader.

The authors frequently refer to the study as providing an indicator of climate change, but never describes what the indicator is. Is the estimated abundance of adults of each species from the spatial GAMs the indicator? Are the environmental parameters themselves the indicator, based on the relationship they display with abundance? And, if so, how are the environmental covariates indicators if the relationship between the species and an individual parameter vary? Similarly, the authors state the use of the investigation in assessing the climate impacts on fisheries, but don’t link the results to fisheries until the final paragraph, and only in relation to brown shrimp. The examination of the relationship between the results and fisheries could be made stronger by including more information on how the analyses could be applied, or what they suggest for fisheries landings.

Finally, the manuscript needs to be thoroughly reviewed and edited by the authors. There are numerous typos (including incorrect spellings, e.g., “Chesapeake” sometimes spelled “Chesepeake”), grammatical errors, and inconsistent tenses throughout the text. The methods section especially needs to be expanded upon and edited for clarity.

Overall, I think the investigation is very interesting and attempts to address a challenging question, the relationship between various drivers and abundance for several species and life stages. I think that with increased attention to clarity for the readers, and addressing the statistical issues, this study will provide valuable information on penaeid shrimp.

Detailed comments are below.

Line 21: The three species are local to the US East Coast and Gulf of Mexico, is it an overstatement to describe them as highly valuable for the US as a whole?

Lines 22-23: What is the significant climate and biogeographic break? Cape Hatteras? This is described in the introduction, but for clarity the authors should consider including it in the abstract as well.

Lines 26-30: The relationships between adult abundances and environmental factors were also explored. The authors conducted more analyses than this sentence makes it seem.

Line 34: Indicating that the adult spawning populations are from previous seasons would help readers understand the difference between the covariate and response variable.

Line 41: Should “percent” be %?

Line 52: It might be valuable to add a sentence or two describing why Cape Hatteras is a significant climate/biogeographic break.

Line 58: Unless commercial shrimp trawling is a new fishery in Pamlico Sound, started because of shifts related to climate change, it should be noted separately, as the sentence is describing the impacts of climate change on Pamlico Sound.

Line 64: Is $22.3 million the revenue from the shrimp fishery (i.e., landings value)? If so, the revenue is not the money brought in to the state.

Line 88-91: Consider listing the objectives in the same order they are presented in the methods and results.

Lines 91-93: This a summary of the results, should it be included in the introduction?

Line 107: Later, P120 is used as the spring survey. For clarity, “spring” should be used in the survey description.

Lines 112-114: Is the P120 survey being limited to 102 fixed sites? Or are there 102 fixed sites total, and the limitation is that the survey data can only support the analysis of juvenile brown shrimp?

Lines 113-114: P120 samples 102 fixed stations annually, but lines 106-107 state that the survey samples twice in the spring (two weeks in May and two weeks in June) – are only some portion of the fixed stations sampled in each month? If so, how is it split and why? Or are all 102 sites sampled each month?

Throughout the description of the P120 methods, it may be useful to have “the P120 survey” interspersed throughout (rather than only “the survey”), for clarity that the entire paragraph is referring only to P120.

Consider limiting P120 data used to 1987-2019 (instead of 1986) for consistency with P195 and all other analyses.

Line 127: The stratification scheme is described in the figure caption, but not in the main text.

Line 128: Clarify that P195 June is considered summer and P195 September is fall.

Line 129: Describing the analyses of “data from 1987 to 2019” implies that P120 is also included in this analysis, when only P195 data are being used.

Lines 129-130: If P120 is capturing the peak abundance of juvenile brown shrimp in June (and May), then is P195 able to adequately capture adult brown shrimp also in June?

Line 135: How are the data expanded to include zeros for any site that didn’t have catch of a particular species if P195 is sampled using a stratified random design? Are the “sites” in P195 the depth strata (or other stratifications)?

Lines 143, 145: What are the months associated with each season?

Line 147: Is CPUE still catch per hectare for SEAMAP? Or is there a different unit for effort?

A map of SEAMAP depicting the strata may be useful (or at least list each of the six strata)

Lines 147-149: It is unclear to me how the North Carolina specific index was calculated. My interpretation of the methods described is that within each stratum, the species-specific CPUE is summed across all sites in a given year and season. Then each of these 6 values are averaged to get a single mean annual log CPUE. But I would interpret that as an overall fisheries-independent index for that season and year, not a North Carolina-specific index.

Line 167: What are the months associated with these seasons?

Lines 172-173: In line 167, 3 seasons are listed (winter, spring, and summer), but these lines only describe winter and spring.

Lines 177-184: I don’t understand the methods in this section. When applying the ClimWin package, is the abundance of a given species the response variable? And, if so, from which survey? If north windstress in December might be important for the following fall white shrimp (abundance?), how is it calculated during the spawning and recruitment periods, if white shrimp spawning and recruitment occurs May-July?

Lines 190-191: Describe the standardization and mean calculations (e.g., average/day then averaged for time period).

Line 192: Were any other broad scale environmental drivers considered? The Gulf Stream was mentioned in the text, perhaps the Gulf Stream Index would be of value.

Line 197: Why was the distribution modeled for the adult shrimp, but not juvenile brown shrimp? And why were spatiotemporal models necessary?

Lines 202-203: This section could benefit by being expanded upon and providing more information on decisions made. E.g., why Tweedie instead of lognormal? Should the reference be Wood 2017, not 2020? How was the value of 1.4 decided upon? Is it the default or through estimation? Simulation? Sensitivity testing?

Equations of the Tweedie distribution would be beneficial – the power parameter is listed in table 2, but is not mentioned in the methods.

Lines 214-215: What is “separate” about the spatial term? Spatial only with no temporal variation?

Great job explaining the differences in the four model types!

How were the GAMs selected?

Lines 221-224: While I think replacing the unrealistic predictions at the edges is a good decision, is using the maximum observed a good assumption? Were the maximum observed catches at the periphery? Or should the unrealistic estimates on the periphery be more similar to surrounding cells, rather than the maximum?

Lines 233-236: If I am interpreting this correctly, three models were fit (May, June, and both CPUE) using different data as the response variable and then compared to one another. This can’t be done as they have different response data. While the CPUE data are understandably easy to incorrectly conflate, as they come from the same survey, comparing models with the three different CPUE indices is akin to comparing the models of the three different shrimp species and selecting only one species that is most described by the considered covariates. This erroneous comparison and selection of the models undermines all conclusions made regarding the environmental factors and juvenile brown shrimp recruitment.

Lines 240-241: This was already described in detail in the data section of the methods, and doesn’t need to be repeated

Lines 252-267: While this section provides important information on model selection, I think it would be much more valuable if included in the individual sections for which it was relevant. While reading through the methods, I had questions regarding model selection and standardization, which were described in this section. Additionally, including everything in a single section, rather than in the relevant sections, makes it unclear which tests or applications were used for which models. There is no description regarding model selection of the GAMs.

Lines 255-256: Why were quadratic terms used for nonlinear relationships rather than smoothers (i.e., GAMs)?

Lines 272-275: I think a description of the model output would be more valuable than a restatement of the methods.

Line 275: Figure 2 is referring only to adult brown shrimp abundance, but is used in the context of all three shrimp species.

Figure 2: Why was 2019 chosen for the spatial maps? Was there something significant about that year. If not, state that 2019 is being used as an example.

Figure 2 caption, 285: List months associated with the seasons.

Figure 2 caption, 288-289: The figure is displaying only adult brown shrimp abundance in the summer and fall in 2019, but the caption references all three species and years.

It would be useful to highlight any major changes in spatial distribution over time, or comparisons between seasons. There isn’t any output from the spatial models referenced in the results, but it was one of the objectives of the investigation, and a main section in the methods.

Consider including all spatial maps for each species, year, and season in the supplementary figures.

Lines 275-276: How does deviance explained visually reflect survey data? Is it the spatial output that reflects survey data?

Table 2 Lists the final GAM chosen for each species and season (which is very useful!), but this could also be briefly described in the results, then reference table 2 (E.g., which species shared model forms).

Lines 277-278: Similar to above, this is a restatement of the methods, rather than a description of the results. The following sentence references Figure 3.

Lines 279-283: The trends in mean survey CPUE and commercial catch data were not described previously. A figure of these trends would be useful.

Figure 3: I think a line plot would be more useful than the bar chart. Consider making the X axes labels begin at the first year and end at the last.

Figure 3 caption: List months associated with seasons.

Table 2 caption: Include scientific names and indicate that these were the chosen models.

Lines 300-310: The authors seem to not be considering the significant interaction between east-west and north-south windstress in the conclusions they are drawing. Because of the interaction, the relationship between one of the windstresses and the CPUE can’t be described without considering the other windstress.

Figure 4: Again, due to the significant interaction between north-south windstress and east-west windstress, the individual relationships of each windstress and CPUE can’t be modeled without including the other windstress.

Table 3 is very useful!

Table 3: List species and scientific names in title/caption.

Figures 4-10 captions: The relationship is between the log CPUE (not CPUE) and the covariates.

Lines 390-392: How was the significant correlation tested?

The results section was very thorough! I don’t think it’s a required change, but the authors could consider providing a brief summary of the results and directing the reader to table 3, rather than describing all variables in the model for each species and season.

Line 434: I’m not sure what is meant by “annual estuarine-dependent species.”

Lines 436-437: I don’t think it has been made clear what metric the authors are suggesting could be used as a bioclimatic indicator. The abundance of adult shrimp?

Lines 437-438: Recruitment was only investigated for brown shrimp.

Line 439: I think it is important to also mention in this paragraph that all models (except adult brown shrimp in the fall) include SEAMAP CPUE or commercial catch as a covariate, which indicates that not only environmental conditions are key in recruitment and abundance of penaeid shrimp, as these are measures of previous spawning stock biomass, and the influence of the environment on survival of penaeid shrimp in coastal waters is outside the scope of this work.

Line 446-448: I think the first half of the sentence should be edited for clarity, but I think it is a valuable conclusion.

Lines 459-462: Great point, really interesting thought!

Line 470-471: This seems like an overstatement outside of the scope of this study, if other alternatives aren’t mentioned. The decline could be related to numerous factors, such as increases in predators, shifts in distribution, etc.

Lines 476-487: While NAO is related to salinity changes (as it is related to many environmental conditions), the explicit comparison between NAO relationships and salinity seems to be oversimplifying NAO. Additionally, after a description of differences in the impacts of salinity, the discussion shifts back to NAO. I think the paragraph would be more clear and fluid if the salinity section was removed from the NAO paragraph.

Lines 488-493: This is a restatement of the results section, which was already described in detail.

Lines 499-513: Really interesting paragraph!

Lines 550-552: While this is an interesting hypothesis, I think it is an overstatement. The negative correlation could be explained by the brown shrimp leaving Pamlico Sound earlier, as the authors suggest. However, the correlation does not explicitly support that hypothesis. It could also be explained by density dependent mortality: increased competition leads to increased mortality (possibly through lack of resources or the foraging arena hypothesis), and therefore lower abundance in the fall.

Lines 562-578: Interesting discussion but the authors should consider reorganizing the paragraph. Similar to other points I have made, the paragraph begins with one topic (pink and white shrimp), then moves on to a second topic (brown shrimp), then returns to the original topic (pink and white shrimp).

Reviewer #2: Summary: This study uses two fishery-independent surveys conducted in North Carolina that span 30 years to examine environmental drivers affecting adult penaeid shrimp (white, pink, and brown shrimp) and recruitment of juvenile brown shrimp using numerical models. Additional coastal data on shrimp abundance were also used as surrogates for spawning stock biomass. Years with higher water temperature, salinity, offshore wind stress, and NAO phase predicted increased abundance of juvenile brown shrimp. Additionally, adult white, pink, and brown shrimp were affected by winter temperatures, wind stress, salinity, the NAO, and index of spawning adults. These results could be used to predict and develop climate-based adaptive management strategies.

Note: According to WoRMS (World Register of Marine Species) the genus for these species has reverted back to Penaeus and Farfantepenaeus and Litopenaeus are no longer recognized (changed as recently as August 2022). Will need to change the names throughout the manuscript.

This is a well written paper that addresses and important fishery and increases our understanding of factors that affect shrimp populations in North Carolina. I think the methods are robust and the data used in the analysis are appropriate for the questions that were asked. A few minor comments and suggestions are below for consideration.

Line-specific comments:

Line 89: Add and ‘s’ at the end of ‘habitat’.

Lines 125-129: I think the first sentence needs to be connected to the subsequent sentences. Perhaps start with the second sentence and combine the two. “The P195 survey ….intercepts shrimp as they grow and migrate towards coastal inlets….”.

Lines 132 – 133: I think a length frequency plot of these data for the three species would be helpful to understand the catch of shrimp. The terms ‘subadult’ and ‘adult’ are subjective and seeing the range of lengths observed in Pamlico Sound could be useful to other investigators from different areas.

Line 161: delete ‘additioanlly’.

Line 199: delete the second appearance of ‘adult’.

Lines 249 - 250: I am curious if you see concordance between the largest P120 shrimp and the smallest P195 shrimp from year to year? Maybe a length frequency plot of shrimp lengths from the two surveys would help me understand differences between the catch. This ties back to the comment for lines 132 – 133.

Line 447: Add a comma after ‘…short-lived species,’.

Line 468: Change ‘parameter’ to ‘factor’.

Lines 612+: This could be its own paragraph and I am torn as to whether is should be. I stumbled a bit while reading it, but leave that to the authors to decide.

Reviewer #3: This research aimed to identify environmental and climate influences on population size of brown, white, and pink shrimp in Pamlico Sound, NC. This is a well-written paper describing original research that is needed to better understand the drivers of penaeid stock dynamics, understand climate change implications for these species, and improve management of these valuable fisheries. Overall, the analyses are well described and presented clearly and the study’s implications are well addressed. I have a few questions and requests for clarification regarding the analyses conducted that I hope will improve the manuscript and support its publication.

My biggest question regarding these analyses is why commercial catch was used solely as an index of spawning stock abundance and why the impacts of the fishery as a potential driver of system dynamics were not considered. Fishery impacts often work in tandem with environmental conditions and climate to drive population changes over time. If effort (e.g., number of trips – line 619 and citation 19) has varied over time, should it not be considered as a factor in the adult models? Unless I misunderstood completely (which is possible!), these models assume the fishery has had no impact on stock dynamics. Line 614 says that “effort in the fishery is influenced by shrimp abundance”, but what if it’s the other way around as in most fisheries? This phenomenon, if real for these stocks, should be better explained and clearly justified. If not, can you demonstrate that including effort would not change your results and interpretation?

Other questions/comments:

1. Lines 135/149 – how did you deal with zero catch in your log(CPUE) models?

2. Line 154 – What does “standardized pounds of shrimp” mean? Standardized by/with what? Please be more specific about how your fishery-dependent index was generated.

3. Line 182 – what about pink shrimp?

4. Line 185+ - include units for flow

5. Line 224 – how might this assumption have impacted your results? Why did you choose spatial GAMs over other spatial interpolation methods?

6. Line 232 – variable should be plural

7. Line 258 – does this mean you then ran two models (one with each, all else the same)? Or did you make a judgement call as to which to include in all future models?

8. Line 279 – you mention CPUE and commercial catch trends but don’t show them. Might be a nice additional set of figures for the reader’s benefit.

9. Line 303 – in the juvenile brown shrimp recruitment model, you use “an index of spawning adult biomass using commercial catch from the previous year”. Why not lagged brown shrimp CPUE from the P195 survey?

10. Line 304 – which index of relative importance was used? There are a few out there.

11. Line 509 – deeper is misspelled

12. One co-author missed a good opportunity to self-cite! I suggest adding a small bit of discussion of Figure 3 trends vs trends in relative abundance in Lee & Rock 2018, addressing if/how/why reported trends differ to give context with relatively recent previous literature.

6. PLOS authors have the option to publish the peer review history of their article (what does this mean?). If published, this will include your full peer review and any attached files.

Reviewer #1: No

Reviewer #2: No

Reviewer #3: No

---

## [Author Response · Author response to Decision Letter 0]

27 Jan 2023

Dear Dr. Ottersen,

We are pleased to submit the revised research article entitled “Environmental and climate variability drive population size of annual penaeid shrimp in a large lagoonal estuary” for consideration of publication in PLOS ONE (Manuscript ID: PONE-D-22-30838).

We are grateful that all three reviewers recognized the importance of our research questions. Reviewer 1 provided an extremely thorough review of our work and they, along with the other two reviewers, raised some very important considerations. We have spent time reviewing all the suggestions made and we agree with and have incorporated nearly all their proposed changes. In the rare case that we have declined to make a suggested edit we have provided our rationale. To address some of the reviewer comments we have added three supplementary figures. We are providing a version of our manuscript with track changes marked as well as a clean version to facilitate easy review. We have responded to all the comments with updated line numbers (which refer to the line numbers for the version without track changes) and modified text in the response. 

Additionally, in response to your questions (1) we have ensured that the manuscript and figure files are in accordance with PLOS ONE’s style requirements, (2) all the data in the manuscript are collected by the North Carolina Division of Fisheries Management and therefore do not require a permit (this is now stated explicitly)--these data are publicly available and are cited in the methods, (3) we have removed the funding statement from the acknowledgements, please update our Funding Statement to read "This research was supported by a grant from North Carolina Sea Grant (R/NCSG-RM-02) to JWM and NH. The funders had no role in study design, data collection and analysis, decision to publish, or preparation of the manuscript." And (4) the maps in Figures 1 and 2 were created with publicly available data using the ‘tidyverse’ package in R and are not copyrighted. This is now specifically stated in the methods.

We appreciate your time and effort in considering our resubmitted manuscript and look forward to your comments.

Sincerely,

Lela S. Schlenker, on behalf of all coauthors

Response to Reviewers

Lela S. Schlenker for the manuscript “Environmental and climate variability drive population size of annual penaeid shrimp in a large lagoonal estuary” for consideration of publication in PLOS ONE (Manuscript ID: PONE-D-22-30838).

Reviewer #1: Please find my review of manuscript ID PONE-D-22-30838 entitled “Environmental and climate variability drive population size…” by Schlenker et al. This investigation explored the relationship between various environmental factors and their influence on the abundance of adult brown, pink, and white shrimp and juvenile brown shrimp in Pamlico Sound. The significant relationship between previous spawning stock sizes and the abundance realized in the fisheries independent trawl surveys lends credence the widely-held conclusion that the spawner-recruit relationship for short-lived species is directly related. They also found interesting associations between environmental factors and abundance of penaeid shrimp. The discussion section of this manuscript was extremely thorough, provided many thoughtful links to the literature, and posited compelling hypotheses. This study provides valuable information on potential drivers of abundance and recruitment of these species. However, I am very concerned about the rigor and validity of the statistical methods employed in this analysis.

My most pressing statistical concern, that needs to be addressed to proceed with publication, is regarding the model of juvenile brown shrimp abundance. If I am properly understanding the authors’ methods, they fit GLMs to the average annual CPUE from May, June, and combined May and June, then compared these three models to one another to determine which GLM had the largest deviance explained. The response data used in each of these three models is different, and therefore, they cannot be compared to one another. If the response data are not shared amongst a set of models, then it is not statistically sound to compare these models and choose the “best” fit. This also means that the conclusions are overstated and misleading. The GLM does not provide information on the relationship between the environmental covariates and brown shrimp recruitment, as suggested by the authors, but instead provides information on the relationship between the environmental factors and CPUE of juvenile brown shrimp in June only. These same environmental factors may not have displayed a relationship with the CPUE of juvenile brown shrimp in May, or the CPUE for the spring. As a reader, it is impossible to know since the results of these models were not reported. I would suggest using the average annual CPUE for the spring (i.e., May and June) if the authors intend to draw conclusions on the overall relationship between the environmental factors and recruitment of juvenile brown shrimp. If I am misunderstanding the methods employed for this analysis, then the methods and results need to be rewritten to clarify that proper statistical procedures were followed.

An additional important revision is regarding the conclusions drawn between windstress and juvenile brown shrimp CPUE in June and adult brown shrimp abundance in the fall. Both of these models included significant interactions between east-west and north-south windstresses. Because of the significant interaction, the relationship between abundance and one windstress cannot be described without including the other windstress. That is, the influence of north-south windstress (as an example) on abundance is dependent upon the east-west windstress. To continue the example, figure 5 demonstrates that lower values of north-south windstress have a positive relationship with juvenile brown shrimp abundance in June, but only when east-west windstress is also low. The authors describe the relationship between abundance and each of the windstresses individually in the text, and display the individual relationships in figures (4 and 7), without the inclusion of the other directional windstress when doing so. The interaction also needs to be considered in the discussion section of the manuscript.

The spatiotemporal modeling section is interesting, but the benefit to using a spatiotemporal model is never described (e.g., is there spatial and/or temporal autocorrelation in the data?). Additionally, there was no spatial modeling of the juvenile brown shrimp, only adults, and it is not explained as to why the juvenile brown shrimp model doesn’t follow the same methods as the adult models. Further, a description as to why the output of the spatial models was summed for use in the GLMs (i.e., so the response variable is on the annual scale for evaluation) would be useful for the reader.

The authors frequently refer to the study as providing an indicator of climate change, but never describes what the indicator is. Is the estimated abundance of adults of each species from the spatial GAMs the indicator? Are the environmental parameters themselves the indicator, based on the relationship they display with abundance? And, if so, how are the environmental covariates indicators if the relationship between the species and an individual parameter vary? Similarly, the authors state the use of the investigation in assessing the climate impacts on fisheries, but don’t link the results to fisheries until the final paragraph, and only in relation to brown shrimp. The examination of the relationship between the results and fisheries could be made stronger by including more information on how the analyses could be applied, or what they suggest for fisheries landings.

Finally, the manuscript needs to be thoroughly reviewed and edited by the authors. There are numerous typos (including incorrect spellings, e.g., “Chesapeake” sometimes spelled “Chesepeake”), grammatical errors, and inconsistent tenses throughout the text. The methods section especially needs to be expanded upon and edited for clarity.

Overall, I think the investigation is very interesting and attempts to address a challenging question, the relationship between various drivers and abundance for several species and life stages. I think that with increased attention to clarity for the readers, and addressing the statistical issues, this study will provide valuable information on penaeid shrimp.

Thank you for your thorough review of our manuscript. We very much appreciate the time that you spent with our work, and we take your concerns about our juvenile brown shrimp model selection, our discussion of windstress interactions, and language around our spatial models seriously. We believe we have addressed your concerns and have responded to each item below. Your attention to detail has improved the manuscript and we are grateful for your review. 

Detailed comments are below.

Line 21: The three species are local to the US East Coast and Gulf of Mexico, is it an overstatement to describe them as highly valuable for the US as a whole?

The statement has been amended

21: economic value to the southeastern United States and Gulf of Mexico

Lines 22-23: What is the significant climate and biogeographic break? Cape Hatteras? This is described in the introduction, but for clarity the authors should consider including it in the abstract as well.

Cape Hatteras has been added to the sentence

22: system that straddles Cape Hatteras, one of the most significant

Lines 26-30: The relationships between adult abundances and environmental factors were also explored. The authors conducted more analyses than this sentence makes it seem.

Edited to read

29-30: the environmental drivers associated with adult shrimp abundance and juvenile brown shrimp

Line 34: Indicating that the adult spawning populations are from previous seasons would help readers understand the difference between the covariate and response variable.

Good point

35: the abundance of spawning adult populations from the previous year on shrimp abundance

Line 41: Should “percent” be %?

This has been changed (line 43)

Line 52: It might be valuable to add a sentence or two describing why Cape Hatteras is a significant climate/biogeographic break.

We agree this is an interesting topic but because it is not the focus of this manuscript, we will leave it as it is currently written with just a brief mention of the importance of Cape Hatteras as a biogeographic break point 

Line 58: Unless commercial shrimp trawling is a new fishery in Pamlico Sound, started because of shifts related to climate change, it should be noted separately, as the sentence is describing the impacts of climate change on Pamlico Sound.

This has been modified

57: Among these changes are increased salinity caused by increasing sea level rise [14], elevated temperatures [15], shifts in species composition [15], eutrophication [16], and increased storm frequency [17]. Additionally, commercial shrimp trawling continues to heavily impact the Pamlico Sound and coastal shelf food web [18, 19].

Line 64: Is $22.3 million the revenue from the shrimp fishery (i.e., landings value)? If so, the revenue is not the money brought in to the state.

Yes, this has been clarified

66: in 2020, the most recent year for which data are available, the landings value of the shrimp harvest was $22.3 million [22].

Line 88-91: Consider listing the objectives in the same order they are presented in the methods and results.

Thank you for this suggestion. 

90: examine: 1) the spatial distribution and abundance of sub-adult and adult penaeid shrimp and 2) the environmental variables that influence the recruitment success of juvenile brown shrimp and the abundance of sub-adult and adult brown, white, and pink shrimp.

Lines 91-93: This a summary of the results, should it be included in the introduction?

Thanks for this comment. We feel that this extremely brief summary of results serves as a “teaser” for more detailed results to come and is appropriate here

Line 107: Later, P120 is used as the spring survey. For clarity, “spring” should be used in the survey description.

Thanks for catching this, we have added this to the initial description

103: Estuarine Trawl Survey (Program 120, hereafter referred to as “P120” or the “spring” survey)

Similarly, for P195

127: Pamlico Sound Trawl Survey (Program 195, hereafter referred to as “P195” or the “summer and fall” survey)

Lines 112-114: Is the P120 survey being limited to 102 fixed sites? Or are there 102 fixed sites total, and the limitation is that the survey data can only support the analysis of juvenile brown shrimp?

Thank you for pointing out that this was confusing. It has been rewritten:

113: Therefore, our analyses of the P120 survey data used only juvenile brown shrimp data. Effort in the survey has varied over time so we opted to use data only from 102 fixed core stations that are sampled in May and again in June within the Pamlico Sound system.

Lines 113-114: P120 samples 102 fixed stations annually, but lines 106-107 state that the survey samples twice in the spring (two weeks in May and two weeks in June) – are only some portion of the fixed stations sampled in each month? If so, how is it split and why? Or are all 102 sites sampled each month?

All 102 sites are sampled each month. See above.

Throughout the description of the P120 methods, it may be useful to have “the P120 survey” interspersed throughout (rather than only “the survey”), for clarity that the entire paragraph is referring only to P120.

Thanks for this suggestion, we have rewritten this paragraph to include only P120 information and the next to only include P195 information and made sure to add in a few specific P120 references in place of “the survey”

Consider limiting P120 data used to 1987-2019 (instead of 1986) for consistency with P195 and all other analyses.

While we agree that this would simplify discussion, we feel it is important to maximize the data available and we are retaining data from 1986 for the juvenile brown shrimp models

Line 127: The stratification scheme is described in the figure caption, but not in the main text.

Good point.

126: Catch data for adult brown, white, and pink shrimp was collected by NCDMF for their Pamlico Sound Trawl Survey (Program 195, hereafter referred to as “P195”) [31]. The P195 trawl survey began in 1987 and occurs each June (summer) and September (fall; mean=52.5 hauls per season; Table 1). The survey (20-minute tows) uses a stratified random design based on depth each season and year (Fig 1) and occurs in the main body (> 2 m) of the Pamlico Sound. The design of the P195 survey is intended to intercept shrimp as they emigrate from shallow nursery habitats and pass through the sound (mean depth ~4.8 m), on their way to coastal inlets.

Line 128: Clarify that P195 June is considered summer and P195 September is fall.

See above 

Line 129: Describing the analyses of “data from 1987 to 2019” implies that P120 is also included in this analysis, when only P195 data are being used.

Thanks

133: Our analyses of P195 data from 1987 to 2019

Lines 129-130: If P120 is capturing the peak abundance of juvenile brown shrimp in June (and May), then is P195 able to adequately capture adult brown shrimp also in June?

This is a good question. Thank you for pointing out that this needed clarification. We have added text and a supplementary figure (Fig S1) showing the length frequency distribution of the various surveys to help clarify.

136: Although both the second sampling of P120 sites and the summer P195 survey both occur in June, the relatively long brown shrimp spawning window and the rapid growth of juvenile brown shrimp means that both surveys are well timed and capture the same cohort of brown shrimp. Additionally, due to survey timing, the P195 survey catches adult and sub-adult shrimp of brown, white, and pink shrimp (Fig S1), but hereafter, for our purposes, shrimp caught in this survey will be referred to as “adult.” 

Line 135: How are the data expanded to include zeros for any site that didn’t have catch of a particular species if P195 is sampled using a stratified random design? Are the “sites” in P195 the depth strata (or other stratifications)?

Thanks for pointing out that this was unclear. The data contains a list of stations where trawls were completed and if there is no catch of any of the three species (P195) or brown shrimp (P120) they are not noted in the dataset. We expanded the dataset to include those as zeros rather than omissions. We added some text to help clarify

143: the dataset was expanded to include zeros for any site (i.e., location where trawling was completed) that did not include catch of a particular species. 

Lines 143, 145: What are the months associated with each season?

Thanks, we’ve added these

153: For each year and season (spring: mid-April to mid-May, summer: mid-July to early August, and fall: early October to mid-November)

Line 147: Is CPUE still catch per hectare for SEAMAP? Or is there a different unit for effort?

Yes

155: the mean log CPUE (individuals per hectare) of adult…

A map of SEAMAP depicting the strata may be useful (or at least list each of the six strata)

We have added the regions of the strata to the text (we are not aware of individual stratum names). We appreciate your interest in a map-level detail but since we are not providing maps for the other predictor variables (i.e., specific buoy locations or stream gauges) we do not feel that this level of detail is necessary.

157: The North Carolina specific index was calculated by averaging the species-specific CPUE for each of the six nearshore SEAMAP strata located between Cape Fear and Cape Hatteras (i.e., the Onslow and Raleigh Bay regions) before determining the mean annual log CPUE across North Carolina strata within representative seasons.

Lines 147-149: It is unclear to me how the North Carolina specific index was calculated. My interpretation of the methods described is that within each stratum, the species-specific CPUE is summed across all sites in a given year and season. Then each of these 6 values are averaged to get a single mean annual log CPUE. But I would interpret that as an overall fisheries-independent index for that season and year, not a North Carolina-specific index.

We refer to this index as a North Carolina-specific index because we are only using North Carolina Strata. There are a total of 24 strata that SEAMAP samples and we used 6 from the NC Coast north of Cape Fear. We have added some additional text to help clarify (see above comment).

Line 167: What are the months associated with these seasons?

Thanks for pointing this out-we have modified the sentence to accurately reflect what seasonal temperature data were used

179: water temperature (winter and spring)

Lines 172-173: In line 167, 3 seasons are listed (winter, spring, and summer), but these lines only describe winter and spring.

See above 

Lines 177-184: I don’t understand the methods in this section. When applying the ClimWin package, is the abundance of a given species the response variable? And, if so, from which survey? If north windstress in December might be important for the following fall white shrimp (abundance?), how is it calculated during the spawning and recruitment periods, if white shrimp spawning and recruitment occurs May-July?

Thanks for pointing out that this was confusing. We used the estimated abundance from the GAM spatial models with the ClimWin package. We’ve added text in two places to clarify. The periods of interest outside of spawning and recruitment periods (e.g., Decemeber for white shrimp) were calculated in addition to the windstress during the spawning and recruitment window and all were evaluated during model fitting.

189: we used the ClimWin package, an approach that uses a sliding window to compare the effect of various time periods of an environmental variable on a chosen response variable (in this case, estimated adult shrimp abundance, see below for details), to identify more specific periods of recruitment impact in our windstress data [36]. This approach highlighted north windstress in the month of April and the previous December as potentially important predictor variables for the estimated abundance of summer brown shrimp and fall white shrimp, respectively, and these windstress values from specific time periods were calculated as described for the spawning and recruitment periods and evaluated in model fitting. This approach did not identify a period of windstress importance for pink shrimp outside of the spawning and recruitment window.

251: This annual estimate was used as the modeled response variable for evaluating environmental predictors of adult shrimp abundance (see below) and for identifying important periods of windstress outside of the spawning and recruitment window with the ClimWin package [36] (see above).

Lines 190-191: Describe the standardization and mean calculations (e.g., average/day then averaged for time period).

Thanks for pointing out this was missing

204: Monthly stream flow data in cubic feet per second were used to calculate the mean annual flow (up to the date the survey began) and the mean flow during the spawning and recruitment period were calculated and standardized to a mean of 0 and a standard deviation of 1 for use in models.

Line 192: Were any other broad scale environmental drivers considered? The Gulf Stream was mentioned in the text, perhaps the Gulf Stream Index would be of value.

All the environmental drivers we considered are described. The Gulf Stream Index is a good idea and one we will consider in the future.

Line 197: Why was the distribution modeled for the adult shrimp, but not juvenile brown shrimp? And why were spatiotemporal models necessary?

Thank you for noting we had not properly explained the need for the GAMS and why they were not used for the juvenile brown shrimp. Text has been added to clarify this

216: These GAMs were used as an approach to account for the spatial autocorrelation inherent in survey data. It should be noted that we used this approach with the P195 survey, which occurs across a continuous section of the Pamlico Sound, but, due to the nature of the P120 sampling, which occurs in discrete tidal creeks spread throughout the Pamlico Sound, we did not consider spatial GAMs appropriate for the P120 juvenile brown shrimp data.

Lines 202-203: This section could benefit by being expanded upon and providing more information on decisions made. E.g., why Tweedie instead of lognormal? Should the reference be Wood 2017, not 2020? How was the value of 1.4 decided upon? Is it the default or through estimation? Simulation? Sensitivity testing?

The Wood reference has been corrected to 2017, thank you for noticing that error. We also added text to provide more information on the Tweedie distribution and the choice of 1.4 as a gamma penalty. We also added a citation for Kim and Gu, 2004 which goes into detail about the choice of 1.4.

222: The Tweedie error distribution is particularly useful for zero-inflated data (such as survey data) and allows for the use of a single model, while the gamma penalty of 1.4 forces each model effective degrees of freedom to count as 1.4 degrees of freedom which increases smoothness and reduces overfitting [39, 40].

Equations of the Tweedie distribution would be beneficial – the power parameter is listed in table 2, but is not mentioned in the methods.

We feel that by including the four model types we evaluated (line 278-283) we are providing sufficient and practical information on the GAMs. The Tweedie power parameter is calculated automatically by the R package and in our estimation, does not need to be described in the methods. If the editor feels that the full equation would enhance the manuscript, we would be willing to include it.

Lines 214-215: What is “separate” about the spatial term? Spatial only with no temporal variation?

Correct-we added text to clarify

236: Model types 2-4 included a spatial term s (longitude, latitude) independent from temporal variation

Great job explaining the differences in the four model types!

Thank you!

How were the GAMs selected?

We had previously attempted to explain all model selection in one final paragraph of the methods section. Based on your note that this was confusing we have added text about GAM selection here. 

241: A model for each species and season was selected using Akaike’s Information Criterion (AIC) [42, 43].

Lines 221-224: While I think replacing the unrealistic predictions at the edges is a good decision, is using the maximum observed a good assumption? Were the maximum observed catches at the periphery? Or should the unrealistic estimates on the periphery be more similar to surrounding cells, rather than the maximum?

This is a valid point. The high predictions from the GAMs did occur at the edges of the prediction grid and were the result of a high observed catch closest to the periphery so those high values were similar to surrounding cells. Additionally, we want to note that the replacement values were the maximum specific to that species, season, and year (not the overall maximum). We have added some text to make those points clear.

246: To reduce unrealistic predictions that rarely occurred at the edges of the projection grid (i.e., spatial extrapolation error based on a nearby high observed catch), predictions that exceeded the maximum observed catch for each species, season, and year were replaced by the respective maximum observed catch value for that specific species, season, and year.

Lines 233-236: If I am interpreting this correctly, three models were fit (May, June, and both CPUE) using different data as the response variable and then compared to one another. This can’t be done as they have different response data. While the CPUE data are understandably easy to incorrectly conflate, as they come from the same survey, comparing models with the three different CPUE indices is akin to comparing the models of the three different shrimp species and selecting only one species that is most described by the considered covariates. This erroneous comparison and selection of the models undermines all conclusions made regarding the environmental factors and juvenile brown shrimp recruitment.

We apologize for the poor phrasing that was used in our submitted manuscript and led to some confusion about our approach here. We are grateful to be able to clarify this and prevent further misconceptions about our approach. The model fitting that was initially described was meant to indicate preliminary data exploration and was not intended to suggest evaluation of model performance using different CPUEs. Our preliminary data exploration suggested that the June CPUE is greater and much more consistent over the timeseries (May Mean CPUE=3.4, May CPUE std dev=1.6; June Mean CPUE=5.6, June CPUE std dev= 0.9) and for this reason we are using the June CPUE in our models. We have rewritten this section to more accurately reflect our approach and rationale for selecting the June CPUE.

260: Because the P120 survey is conducted in May and June (Table 1), we evaluated the CPUE data from both months and found that the CPUE from June was greater and much more consistent over the timeseries than the CPUE from May. For that reason, we opted to develop these models using only the log June CPUE.

Lines 240-241: This was already described in detail in the data section of the methods, and doesn’t need to be repeated

This sentence has been deleted

Lines 252-267: While this section provides important information on model selection, I think it would be much more valuable if included in the individual sections for which it was relevant. While reading through the methods, I had questions regarding model selection and standardization, which were described in this section. Additionally, including everything in a single section, rather than in the relevant sections, makes it unclear which tests or applications were used for which models. There is no description regarding model selection of the GAMs.

Agreed, we’ve modified this accordingly see line 241 and paragraph beginning on 277

Lines 255-256: Why were quadratic terms used for nonlinear relationships rather than smoothers (i.e., GAMs)?

We updated this sentence to be more clear

279: Quadratic terms were evaluated in GLMs for variables where data visualization suggested nonlinear relationships.

Lines 272-275: I think a description of the model output would be more valuable than a restatement of the methods.

Agreed. It now reads

297: The spatial distribution of adult shrimp within the Pamlico Sound (1987-2019) varied on an annual basis and appeared to accurately reflect survey catch data (Table 2). The deviance explained by the GAMs varied by species and season but ranged from 46 to 74 percent.

Line 275: Figure 2 is referring only to adult brown shrimp abundance, but is used in the context of all three shrimp species.

Thanks for pointing this out

303: distributions within the Pamlico Sound (e.g., brown shrimp in 2019, Figure 2)

Figure 2: Why was 2019 chosen for the spatial maps? Was there something significant about that year. If not, state that 2019 is being used as an example.

See above and edits to Figure 2 caption

309: Figure 2. The predicted spatial distribution of adult brown shrimp (Farfantepenaeus aztecus) in Pamlico Sound, North Carolina in the summer and fall of 2019. This year and species were selected as an example. The colored region on the map shows the extent of the trawl survey area and the colors depict the predicted log catch per unit effort of adult brown shrimp using a generalized additive modeling approach. The spatial distribution of each species (brown, white, and pink) of shrimp was estimated for each year and season (1987-2019; summer and fall).

Figure 2 caption, 285: List months associated with the seasons.

See above

Figure 2 caption, 288-289: The figure is displaying only adult brown shrimp abundance in the summer and fall in 2019, but the caption references all three species and years.

This is correct. We hope that by using your suggestion of more clearly stating that that brown shrimp in 2019 were selected as an example that it is apparent in the caption that we are simply referencing that this approach has been used for the other species.

It would be useful to highlight any major changes in spatial distribution over time, or comparisons between seasons. There isn’t any output from the spatial models referenced in the results, but it was one of the objectives of the investigation, and a main section in the methods.

In the future we plan to do a more in-depth analysis of spatial trends. For now, we feel that a qualitative assessment is not adequate, and anything further is beyond the scope of the present work. However, the spatial models were integral to our study as they provided the annual estimates of adult shrimp species.

Consider including all spatial maps for each species, year, and season in the supplementary figures.

Thanks for this suggestion. These maps would be an additional 99 pages of figures (33 years and 3 species). This is something we could do if the editor feels like it is appropriate. 

Lines 275-276: How does deviance explained visually reflect survey data? Is it the spatial output that reflects survey data?

We updated this to provide greater clarity.

297: The spatial distribution of adult shrimp within the Pamlico Sound (1987-2019) varied on an annual basis and appeared to accurately reflect survey catch data (Table 2). The deviance explained by the GAMs varied by species and season but ranged from 46 to 74 percent.

Table 2 Lists the final GAM chosen for each species and season (which is very useful!), but this could also be briefly described in the results, then reference table 2 (E.g., which species shared model forms).

Thanks for this idea.

299: Fall and summer brown shrimp shared the same model formulation as did fall white shrimp and summer pink shrimp, while fall pink shrimp shared the same construction as for brown shrimp but used a higher k value (Table 2).

Lines 277-278: Similar to above, this is a restatement of the methods, rather than a description of the results. The following sentence references Figure 3.

Thank you for pointing this out-this redundancy has been deleted.

Lines 279-283: The trends in mean survey CPUE and commercial catch data were not described previously. A figure of these trends would be useful.

Thanks for this comment, we’ve added a supplemental section with those figures 

304: Much like mean survey CPUE and commercial catch data (Figure S2, S3)…

Figure 3: I think a line plot would be more useful than the bar chart. Consider making the X axes labels begin at the first year and end at the last.

Thanks for this suggestion-we agree. See updated Figure 3

Figure 3 caption: List months associated with seasons.

316: Figure 3. Predicted abundance (millions) of adult brown (Penaeus aztecus), white (P. setiferus), and pink shrimp (P. duorarum) in Pamlico Sound, North Carolina from 1987-2019 for summer (June) and fall (September) using generalized additive spatial models. 

Table 2 caption: Include scientific names and indicate that these were the chosen models.

320: Table 2. Parameters of chosen generalized additive models estimating the spatial distribution of adult brown (Penaeus aztecus), white (P. setiferus), and pink (P. duorarum) shrimp.

Lines 300-310: The authors seem to not be considering the significant interaction between east-west and north-south windstress in the conclusions they are drawing. Because of the interaction, the relationship between one of the windstresses and the CPUE can’t be described without considering the other windstress.

We take your point however the model does also include the individual (east-west and north-south) windstress components (as shown in Table 3) in addition to the interaction and the index of relative importance does highlight the contribution of the east-west windstress. We therefore feel it is correct to show these individual components graphically (per your comment below) and the interaction as we have done in Figure 5. We feel we have stressed the importance of the interaction while still mentioning the individual components and that this is appropriate.

327: This model used the June CPUE of juvenile brown shrimp and included in situ mean salinity and temperature, the interaction between north-south and east-west windstress during spawning and recruitment, NAO phase during the spawning and recruitment period, and an index of spawning adult biomass using commercial catch from the previous year (Fig 4a-f). Using an index of relative importance, salinity, temperature, east-west windstress, and NAO phase were the most important predictors of recruitment (Fig 4g). All variables included in the best model had a positive correlation with the CPUE of juvenile brown shrimp except for east-west windstress (i.e., an offshore or west wind was associated with higher juvenile brown shrimp abundance). The significant interaction between east-west and north-south windstress demonstrated that a southwest-wind during the spawning and recruitment time period was associated with increased juvenile brown shrimp in the P120 survey (Fig 5a).

Figure 4: Again, due to the significant interaction between north-south windstress and east-west windstress, the individual relationships of each windstress and CPUE can’t be modeled without including the other windstress.

See above response, but we do agree this should be mentioned in the Figure 4 and 7 captions and we have added text to that effect.

344: Fig 4….. It should be noted that panels c and e show directional windstress components but see Fig 5a for their interaction.

And 

405: Fig 7…. . It should be noted that panels b and d show directional windstress components but see Fig 5b for their interaction.

Table 3 is very useful!

Glad you like it!

Table 3: List species and scientific names in title/caption.

Thanks for pointing out this omission

361: Table 3. Variables included for best fit generalized linear models predicting the abundance of juvenile and adult brown (Penaeus aztecus), white (P. setiferus), and pink (P. duorarum) shrimp species.

Figures 4-10 captions: The relationship is between the log CPUE (not CPUE) and the covariates.

Thank you, this has been corrected for each caption

Lines 390-392: How was the significant correlation tested?

Thanks, we’ve specified that in the text now

416: winter temperature (linear regression, p<0.01; estimate=0.40; standard error=0.14; t value=2.96, R2=0.21)

The results section was very thorough! I don’t think it’s a required change, but the authors could consider providing a brief summary of the results and directing the reader to table 3, rather than describing all variables in the model for each species and season.

Thank you for the kind note. We appreciate the suggestion but are opting to leave in the longer text description of the variables.

Line 434: I’m not sure what is meant by “annual estuarine-dependent species.”

458: As an estuarine-dependent species with an annual life cycle

Lines 436-437: I don’t think it has been made clear what metric the authors are suggesting could be used as a bioclimatic indicator. The abundance of adult shrimp?

Thanks for pointing out this was unclear

458: the relationship between penaeid shrimp abundance and environmental conditions is tightly linked and adult shrimp abundance may serve as an important bioclimatic indicator that will prove increasingly important to evaluate the impact of climate change on coastal ecosystems

Lines 437-438: Recruitment was only investigated for brown shrimp.

461: The findings from our analyses indicate that the successful recruitment of juvenile brown shrimp and abundance of adult penaeid shrimp in North Carolina

Line 439: I think it is important to also mention in this paragraph that all models (except adult brown shrimp in the fall) include SEAMAP CPUE or commercial catch as a covariate, which indicates that not only environmental conditions are key in recruitment and abundance of penaeid shrimp, as these are measures of previous spawning stock biomass, and the influence of the environment on survival of penaeid shrimp in coastal waters is outside the scope of this work.

We agree and we’ve added a sentence to make that point clear

464: Additionally, nearly all our models included a measure of spawning stock biomass from the previous season, illustrating the important connection between abundant spawners and a strong recruitment class.

Line 446-448: I think the first half of the sentence should be edited for clarity, but I think it is a valuable conclusion.

Good suggestion, thank you

475: Successful recruitment of juveniles to a population frequently plays a key role in population dynamics; however, in the case of short-lived species, the recruiting year class contributes a significant share of the landings and a juvenile recruitment index can be useful for determining total allowable catches [51].

Lines 459-462: Great point, really interesting thought!

Thank you

Line 470-471: This seems like an overstatement outside of the scope of this study, if other alternatives aren’t mentioned. The decline could be related to numerous factors, such as increases in predators, shifts in distribution, etc.

Agreed. We have added the following

499: The increase in white shrimp in the Pamlico Sound that has occurred over approximately the same period, could suggest interspecific competition is contributing to the decline of pink shrimp. However, it should be noted that many other factors not included in our models such as loss of seagrass [53], predation pressure, or shifts in distribution outside of Pamlico Sound could also explain declining numbers of pink shrimp.

Lines 476-487: While NAO is related to salinity changes (as it is related to many environmental conditions), the explicit comparison between NAO relationships and salinity seems to be oversimplifying NAO. Additionally, after a description of differences in the impacts of salinity, the discussion shifts back to NAO. I think the paragraph would be more clear and fluid if the salinity section was removed from the NAO paragraph.

We agree and have created two separate paragraphs. The new salinity focused paragraph: 

514: For estuarine-dependent species such as penaeid shrimp, variations in salinity throughout ontogeny can represent a highly dynamic environmental variable. High salinity had a positive effect on abundance for juvenile brown and one-year old summer pink shrimp while fall pink shrimp were more abundant with increased river volume (lower salinity). However, the relationship between fall pink shrimp and river volume was weaker than the relationship between salinity and summer pink shrimp. As mentioned previously, summer pink shrimp are age-one and have overwintered in the Pamlico Sound [45], while pink shrimp caught in the fall have recently recruited and it is not unexpected that they would show ontogenetic differences in their response to environmental variables.

Lines 488-493: This is a restatement of the results section, which was already described in detail.

Agreed, we deleted the redundant text

Lines 499-513: Really interesting paragraph!

Thank you

Lines 550-552: While this is an interesting hypothesis, I think it is an overstatement. The negative correlation could be explained by the brown shrimp leaving Pamlico Sound earlier, as the authors suggest. However, the correlation does not explicitly support that hypothesis. It could also be explained by density dependent mortality: increased competition leads to increased mortality (possibly through lack of resources or the foraging arena hypothesis), and therefore lower abundance in the fall.

Thanks for this important point, we added text to clarify 

580: The negative correlation between increased early recruitment and decreased fall abundance suggests that when brown shrimp recruit in high numbers to the Pamlico Sound in May they typically grow quickly and leave the sound before the September P195 survey occurs. Or, alternatively, that density dependent mortality or heightened interspecific competition occurs and results in decreased fall abundance.

Lines 562-578: Interesting discussion but the authors should consider reorganizing the paragraph. Similar to other points I have made, the paragraph begins with one topic (pink and white shrimp), then moves on to a second topic (brown shrimp), then returns to the original topic (pink and white shrimp).

Thanks for this suggestion. We have reorganized the paragraph starting on line 594 for better flow.

Reviewer #2: Summary: This study uses two fishery-independent surveys conducted in North Carolina that span 30 years to examine environmental drivers affecting adult penaeid shrimp (white, pink, and brown shrimp) and recruitment of juvenile brown shrimp using numerical models. Additional coastal data on shrimp abundance were also used as surrogates for spawning stock biomass. Years with higher water temperature, salinity, offshore wind stress, and NAO phase predicted increased abundance of juvenile brown shrimp. Additionally, adult white, pink, and brown shrimp were affected by winter temperatures, wind stress, salinity, the NAO, and index of spawning adults. These results could be used to predict and develop climate-based adaptive management strategies.

Thank you for the time you spent reviewing our manuscript. We have responded to each of your suggestions below.

Note: According to WoRMS (World Register of Marine Species) the genus for these species has reverted back to Penaeus and Farfantepenaeus and Litopenaeus are no longer recognized (changed as recently as August 2022). Will need to change the names throughout the manuscript.

This is a well written paper that addresses and important fishery and increases our understanding of factors that affect shrimp populations in North Carolina. I think the methods are robust and the data used in the analysis are appropriate for the questions that were asked. A few minor comments and suggestions are below for consideration.

Thank you for letting us know about the genus change (this has been updated) and for your thoughtful and helpful comments on the manuscript. 

Line-specific comments:

Line 89: Add and ‘s’ at the end of ‘habitat’. 

Done

Lines 125-129: I think the first sentence needs to be connected to the subsequent sentences. Perhaps start with the second sentence and combine the two. “The P195 survey ….intercepts shrimp as they grow and migrate towards coastal inlets….”.

Good suggestion, it now reads: 

128: The P195 trawl survey began in 1987 and occurs each June (summer) and September (fall; mean=52.5 hauls per season; Table 1). The survey employs 20-minute tows in a stratified random design based on depth each season and year and occurs in the main body (> 2 m) of the Pamlico Sound (Fig 1). The design of the P195 survey is intended to intercept shrimp as they emigrate from shallow nursery habitats and pass through the sound (mean depth ~4.8 m), on their way to coastal inlets.

Lines 132 – 133: I think a length frequency plot of these data for the three species would be helpful to understand the catch of shrimp. The terms ‘subadult’ and ‘adult’ are subjective and seeing the range of lengths observed in Pamlico Sound could be useful to other investigators from different areas.

You are correct that the terms adult and subadult are subjective. Based on this comment and the one below we added a figure showing the length frequency distributions for each species and survey (Fig S1)

Line 161: delete ‘additioanlly’.

Done

Line 199: delete the second appearance of ‘adult’.

Done

Lines 249 - 250: I am curious if you see concordance between the largest P120 shrimp and the smallest P195 shrimp from year to year? Maybe a length frequency plot of shrimp lengths from the two surveys would help me understand differences between the catch. This ties back to the comment for lines 132 – 133.

Within a species there is typically some overlap between the largest individuals caught in P120 and the smallest caught in P195. We hope that the inclusion of Fig S1 which shows the length frequency distributions for the three species in the two surveys will help clarify.

Line 447: Add a comma after ‘…short-lived species,’.

Done

Line 468: Change ‘parameter’ to ‘factor’.

Done

Lines 612+: This could be its own paragraph and I am torn as to whether is should be. I stumbled a bit while reading it, but leave that to the authors to decide.

Thank you for this suggestion-we opted to make this a new paragraph.

Reviewer #3: This research aimed to identify environmental and climate influences on population size of brown, white, and pink shrimp in Pamlico Sound, NC. This is a well-written paper describing original research that is needed to better understand the drivers of penaeid stock dynamics, understand climate change implications for these species, and improve management of these valuable fisheries. Overall, the analyses are well described and presented clearly and the study’s implications are well addressed. I have a few questions and requests for clarification regarding the analyses conducted that I hope will improve the manuscript and support its publication.

My biggest question regarding these analyses is why commercial catch was used solely as an index of spawning stock abundance and why the impacts of the fishery as a potential driver of system dynamics were not considered. Fishery impacts often work in tandem with environmental conditions and climate to drive population changes over time. If effort (e.g., number of trips – line 619 and citation 19) has varied over time, should it not be considered as a factor in the adult models? Unless I misunderstood completely (which is possible!), these models assume the fishery has had no impact on stock dynamics. Line 614 says that “effort in the fishery is influenced by shrimp abundance”, but what if it’s the other way around as in most fisheries? This phenomenon, if real for these stocks, should be better explained and clearly justified. If not, can you demonstrate that including effort would not change your results and interpretation?

Thanks for these helpful comments and the time you spent reviewing our manuscript. With regards to commercial catch and its use in models, it seems as though there is a misunderstanding. We did use commercial catch or the SEAMAP CPUE (both of which we assumed to be an index of spawning stock abundance) in model fitting and one of these parameters is in every adult model except for adult brown shrimp in the fall (see Table 3 for clarification). When including these parameters in the models we were unsure of whether a higher index of commercial fishing or the SEAMAP CPUE would predict decreased or increased abundance but across models where these were included, increased spawning stock (whether commercial catch index or SEAMAP CPUE) predicted higher abundance. These results suggest that in this fishery, higher catch is proportional to overall greater abundance and spawning stock biomass. We hope this explanation clarifies the confusion and we will look for places in the text to make sure that this is abundantly clear.

Other questions/comments:

1. Lines 135/149 – how did you deal with zero catch in your log(CPUE) models?

Thanks for pointing out that this was omitted.

Line 277: A value of 1 was added to catches and estimated abundances before values were logged to adjust for values of zero.

2. Line 154 – What does “standardized pounds of shrimp” mean? Standardized by/with what? Please be more specific about how your fishery-dependent index was generated.

Thanks for letting us know that this was unclear. It had previously been described on line 272 (Variables on vastly different scales (e.g., river volume, commercial catch) were standardized using the expression: x-mean (x) / standard deviation (x).) but based on your comment we added a description immediately following the description of the variable. 

165: indices (standardized pounds of shrimp, i.e., a mean of 0 and a standard deviation of 1) from within the Pamlico Sound were calculated and lagged one year. 

3. Line 182 – what about pink shrimp?

This particular analysis did not highlight a period of importance outside of the spawning/recruitment window for pink shrimp, but to make sure this is clear we added the following:

196: This approach did not identify a period of windstress importance for pink shrimp outside of the spawning and recruitment window.

4. Line 185+ - include units for flow

Units have been added

204: Monthly stream flow data in cubic feet per second were used to calculate the mean annual flow (up to the date the survey began) and the mean flow during the spawning and recruitment period were calculated and standardized to a mean of 0 and a standard deviation of 1 for use in models.

5. Line 224 – how might this assumption have impacted your results? Why did you choose spatial GAMs over other spatial interpolation methods?

High predictions from the GAMs did occur at the edges of the prediction grid and were the result of high observed catch or catches closest to the periphery, so those high prediction values were similar to surrounding cells. We feel that our assumption of replacing them with the maximum value specific to that species, season, and year (not the overall maximum) is reasonable. We have added some text to make those points clearer. We also added text to note that spatial GAMs are an accepted methodology to account for spatial autocorrelation inherent to survey data.

246: To reduce unrealistic predictions that rarely occurred at the edges of the projection grid (i.e., spatial extrapolation error based on a nearby high observed catch), predictions that exceeded the maximum observed catch for each species, season, and year were replaced by the respective maximum observed catch value for that specific species, season, and year.

216: These GAMs were used as an approach to account for the spatial autocorrelation inherent in survey data. It should be noted that we used this approach with the P195 survey, which occurs across a continuous section of the Pamlico Sound, but, due to the nature of the P120 sampling, which occurs in discrete tidal creeks spread throughout the Pamlico Sound, we did not consider spatial GAMs appropriate for the P120 juvenile brown shrimp data.

6. Line 232 – variable should be plural

Thanks for catching that!

7. Line 258 – does this mean you then ran two models (one with each, all else the same)? Or did you make a judgement call as to which to include in all future models?

Thank you for pointing out this was unclear. We modified the sentence, it now reads:

281: Collinearity of terms was evaluated using the car package [48] during model fitting and variables with a variance inflation factor greater than three were not included together in the same model and our best judgement was used in determining which variable was more appropriate.

8. Line 279 – you mention CPUE and commercial catch trends but don’t show them. Might be a nice additional set of figures for the reader’s benefit.

We agree that these data are interesting, and we have added them as Supplementary Figure 2 and 3

9. Line 303 – in the juvenile brown shrimp recruitment model, you use “an index of spawning adult biomass using commercial catch from the previous year”. Why not lagged brown shrimp CPUE from the P195 survey?

Thanks for this note. We did evaluate this idea during preliminary model fitting and found that the commercial catch or SEAMAP catch generally was a better fit.

10. Line 304 – which index of relative importance was used? There are a few out there.

This is detailed lines 289-292.

To examine the relative importance of each predictor variable in the best fitted model, all possible sub-model combinations of the final variable set were fitted and AICc scores were calculated for each [47]. From this suite of models, the Akaike weights that included each variable were summed and plotted.

11. Line 509 – deeper is misspelled

Thank you for catching that

12. One co-author missed a good opportunity to self-cite! I suggest adding a small bit of discussion of Figure 3 trends vs trends in relative abundance in Lee & Rock 2018, addressing if/how/why reported trends differ to give context with relatively recent previous literature.

Thank you for this suggestion. Lee & Rock 2018 specifically discusses the importance of evaluating persistence for fixed station sampling designs and evaluates the performance of the P120 survey. Figure 3 shows estimated abundance based on our spatial GAMS which uses the P195 survey data (a stratified random sampling design). So, while we don’t feel it is appropriate to add this citation to the discussion around Figure 3 results, we have added it to the discussion of the juvenile results.

474: Additionally, given that catches of juvenile brown shrimp at the P120 fixed stations are inconsistent from year to year [49], this illustrates the large role of environmental variability and spawning stock biomass in successful recruitment.

---

## [Decision Letter · Decision Letter 1]

14 Apr 2023

PONE-D-22-30838R1Environmental and climate variability drive population size of annual penaeid shrimp in a large lagoonal estuaryPLOS ONE

Dear Dr. Schlenker,

Thank you for submitting your manuscript to PLOS ONE. After careful consideration, we feel that it has merit but does not fully meet PLOS ONE’s publication criteria as it currently stands. Therefore, we invite you to submit a revised version of the manuscript that addresses the one remaining point of rev. 3. Please also see my related comment further down.

We look forward to receiving your revised manuscript.

Kind regards,

Geir Ottersen

Academic Editor

PLOS ONE

Journal Requirements:

Additional Editor Comments:

As the evaluations from the three reviewers show you are very nearly there. However, ref 3 points to one issue that you should look further into before I accept for publication. Further to that point you could check Stratton et al. 2019 Fisheries Oceanography.28:487–504. There they analyzed relative abundance of fish and shellfish species in SEAMAP data and found that in the southeastern US, “for the assemblage as a whole, fishing effects predominated over climate effects. In particular, changes in trawling effort (#trips) within the penaeid shrimp fishery governed abundance trends…changes in trawling effort within the penaeid shrimp fishery governed abundance trends of bony fishes, invertebrates, and elasmobranchs, a likely result of temporal changes in bycatch mortality.”

Reviewers' comments:

Reviewer's Responses to Questions

**Comments to the Author**

1. If the authors have adequately addressed your comments raised in a previous round of review and you feel that this manuscript is now acceptable for publication, you may indicate that here to bypass the “Comments to the Author” section, enter your conflict of interest statement in the “Confidential to Editor” section, and submit your "Accept" recommendation.

Reviewer #1: All comments have been addressed

Reviewer #2: All comments have been addressed

Reviewer #3: (No Response)

2. Is the manuscript technically sound, and do the data support the conclusions?

Reviewer #1: Yes

Reviewer #2: (No Response)

Reviewer #3: Partly

3. Has the statistical analysis been performed appropriately and rigorously? 

Reviewer #1: Yes

Reviewer #2: (No Response)

Reviewer #3: Yes

4. Have the authors made all data underlying the findings in their manuscript fully available?

Reviewer #1: Yes

Reviewer #2: (No Response)

Reviewer #3: Yes

5. Is the manuscript presented in an intelligible fashion and written in standard English?

Reviewer #1: Yes

Reviewer #2: (No Response)

Reviewer #3: Yes

6. Review Comments to the Author

Reviewer #1: (No Response)

Reviewer #2: (No Response)

Reviewer #3: I thank the authors for their thorough response to this lengthy set of reviews. However, my one, primary concern with the study remains. As I understand it, goal #1 of this study was to be able to model “the spatial distribution and abundance of sub-adult and adult penaeid shrimp”. I am still wondering if fishery effort (e.g., #trips) could be an even better explanatory variable for spatial patterns in adult shrimp abundance than catch.

The authors’ response indicated that commercial catch and SEAMAP CPUE were considered as factors in their models (e.g. Lines 150 and 164). But SEAMAP CPUE is an index of abundance and catch alone can be influenced by a lot of things (it’s often hard to say what it’s a measure of). The authors went on to say that “in this fishery, higher catch is proportional to overall greater abundance and spawning stock biomass”, which they have demonstrated. But can you better predict spatiotemporal patterns in abundance with effort than with catch, especially given shrimp trawl fishing effort has been shown to be highly influential (more so than environmental variables) in predicting SEAMAP relative abundance for many southeastern species due to the bycatch impacts of this fishery.

7. PLOS authors have the option to publish the peer review history of their article (what does this mean?). If published, this will include your full peer review and any attached files.

Reviewer #1: No

Reviewer #2: No

Reviewer #3: No

---

## [Author Response · Author response to Decision Letter 1]

17 Apr 2023

From Dr. Ottersen:

As the evaluations from the three reviewers show you are very nearly there. However, ref 3 points to one issue that you should look further into before I accept for publication. Further to that point you could check Stratton et al. 2019 Fisheries Oceanography.28:487–504. There they analyzed relative abundance of fish and shellfish species in SEAMAP data and found that in the southeastern US, “for the assemblage as a whole, fishing effects predominated over climate effects. In particular, changes in trawling effort (#trips) within the penaeid shrimp fishery governed abundance trends…changes in trawling effort within the penaeid shrimp fishery governed abundance trends of bony fishes, invertebrates, and elasmobranchs, a likely result of temporal changes in bycatch mortality.”

Thank you Dr. Ottersen for your help in improving our manuscript. As both you and reviewer three noted, our models use commercial shrimp catch and the SEAMAP CPUE as abundance indicators of shrimp in Pamlico Sound, NC. We feel that this is a stronger approach than using commercial shrimp effort (i.e., number of trips). We are familiar with Dr. Stratton’s paper, and it is cited in our manuscript. Stratton et al. 2019 showed that many species of finfish, elasmobranchs, and invertebrates in the US Southeast were significantly affected by effort in the shrimp fishery. However, this is primarily due to the fact that these species are caught as bycatch in the shrimp fishery. Importantly, shrimp are not a bycatch species (i.e., they are not discarded) and in Stratton et al.’s paper, shrimp abundance was not significantly related to commercial shrimping effort (Stratton et al., 2019, Table 4). Stratton et al. also used effort data because there are no reported bycatch data, and presumably, if estimates of bycatch were available those data would have been an even better predictor than effort data for his study. This suggests to us that adding effort in our models would not improve our model fit. Additionally, a long-standing issue in North Carolina is that effort in the shrimp fishery is poorly resolved. Because shrimp are not a bycatch species in the shrimp fishery, we feel strongly that commercial catch data and the fisheries independent SEAMAP CPUE are much better indicators of abundance and our data support that these variables are good predictors of shrimp abundance. Our models resolve 37-81% (mean: 64%) of the variation in the data indicating that they do a very good job of describing the patterns in abundance data. Thank you again for your care and attention to our manuscript and for considering our response.

Reviewer #3: I thank the authors for their thorough response to this lengthy set of reviews. However, my one, primary concern with the study remains. As I understand it, goal #1 of this study was to be able to model “the spatial distribution and abundance of sub-adult and adult penaeid shrimp”. I am still wondering if fishery effort (e.g., #trips) could be an even better explanatory variable for spatial patterns in adult shrimp abundance than catch.

The authors’ response indicated that commercial catch and SEAMAP CPUE were considered as factors in their models (e.g. Lines 150 and 164). But SEAMAP CPUE is an index of abundance and catch alone can be influenced by a lot of things (it’s often hard to say what it’s a measure of). The authors went on to say that “in this fishery, higher catch is proportional to overall greater abundance and spawning stock biomass”, which they have demonstrated. But can you better predict spatiotemporal patterns in abundance with effort than with catch, especially given shrimp trawl fishing effort has been shown to be highly influential (more so than environmental variables) in predicting SEAMAP relative abundance for many southeastern species due to the bycatch impacts of this fishery.

Thank you for your help in improving our manuscript and for the thoroughness of your reviews. You are absolutely correct that shrimp trawl effort has been shown to be a very important predictor variable, even outweighing climate data, in an important study by Stratton et al. 2019. We are familiar with Dr. Stratton’s paper, and it is cited in our manuscript. However, for several reasons we feel strongly that commercial catch of shrimp and SEAMAP CPUE of shrimp are more appropriate predictor variables than commercial shrimp effort in our models examining the population trends of shrimp in Pamlico Sound, North Carolina. 

As you note, we're using catch data, lagged one year, as an indicator of spawning stock abundance. This assumes that commercial catch might be related to abundance, which is highly variable from year to year. Because catch has biases associated with effort, we are also using SEAMAP indices as an independent abundance indicator. When included in the models (through model selection criteria), commercial catch was positively related to the following year's recruitment, on all accounts, suggesting that increased catch indicated higher abundance (see Figures 4 and 9). If fisheries effects were negatively impacting recruitment, we'd expect a negative relationship, which we do not see.

While a number of species examined in Stratton et al. 2019 were influenced by shrimp trawl effort, shrimp were not among these species. For species that did show such a relationship, it was primarily due to the fact that these were species caught as bycatch in the shrimp fishery. Importantly, shrimp are not a bycatch species (i.e., they are not discarded) and in Stratton et al.’s paper, shrimp abundance was not significantly related to commercial shrimping effort (Stratton et al., 2019, Table 4). Stratton et al. also used effort data because there are no reported bycatch data, and presumably, if estimates of bycatch were available those data would have been an even better predictor than effort data for his study. This suggests to us that adding effort in our models would not improve our model fit. Additionally, a long-standing issue in North Carolina is that effort in the shrimp fishery is poorly resolved. 

Because shrimp are not a bycatch species in the shrimp fishery, we feel strongly that commercial catch data and the fisheries independent SEAMAP CPUE are superior indicators of abundance compared to effort. Most importantly, our results show that both SEAMAP CPUE and commercial catch data are good predictors of shrimp abundance. Our models resolve 37-81% (mean: 64%) of the variation in the data indicating that they do a very good job of describing the patterns in abundance data.

We are thankful to have had such attention to our manuscript from yourself and the other reviewers and are very appreciative of your time and effort. Thank you for helping us to improve our manuscript!

---

## [Editor Report · Decision Letter 2]

25 Apr 2023

Environmental and climate variability drive population size of annual penaeid shrimp in a large lagoonal estuary

PONE-D-22-30838R2

Dear Dr. Schlenker,

We’re pleased to inform you that your manuscript has been judged scientifically suitable for publication and will be formally accepted for publication once it meets any outstanding technical requirements. You have provided a good reply to the final enquiries by one of the reviewers.    

Kind regards,

Geir Ottersen

Academic Editor

PLOS ONE

---

## [Editor Report · Acceptance letter]

5 May 2023

PONE-D-22-30838R2 

Environmental and climate variability drive population size of annual penaeid shrimp in a large lagoonal estuary 

Dear Dr. Schlenker:

I'm pleased to inform you that your manuscript has been deemed suitable for publication in PLOS ONE. Congratulations! Your manuscript is now with our production department. 

Kind regards, 

on behalf of

Dr. Geir Ottersen 

Academic Editor

PLOS ONE